# Non-Asymptotic Error Bounds for Bidirectional GANs

**Shiao Liu**
Department of Statistics and Actuarial Science, University of Iowa
Iowa City, IA 52242, USA
`shiao-liu@uiowa.edu`

**Yunfei Yang** *
Department of Mathematics, The Hong Kong University of Science and Technology
Clear Water Bay, Hong Kong, China
`yyangdc@connect.ust.hk`

**Jian Huang** *
Department of Statistics and Actuarial Science, University of Iowa
Iowa City, IA 52242, USA
`jian-huang@uiowa.edu`

**Yuling Jiao***
School of Mathematics and Statistics, Wuhan University
Wuhan, Hubei, China 430072
`yulingjiaomath@whu.edu.cn`

**Yang Wang**
Department of Mathematics, The Hong Kong University of Science and Technology
Clear Water Bay, Hong Kong, China
`yangwang@ust.hk`

## Abstract

We derive nearly sharp bounds for the bidirectional GAN (BiGAN) estimation error under the Dudley distance between the latent joint distribution and the data joint distribution with appropriately specified architecture of the neural networks used in the model. To the best of our knowledge, this is the first theoretical guarantee for the bidirectional GAN learning approach. An appealing feature of our results is that they do not assume the reference and the data distributions to have the same dimensions or these distributions to have bounded support. These assumptions are commonly assumed in the existing convergence analysis of the unidirectional GANs but may not be satisfied in practice. Our results are also applicable to the Wasserstein bidirectional GAN if the target distribution is assumed to have a bounded support. To prove these results, we construct neural network functions that push forward an empirical distribution to another arbitrary empirical distribution on a possibly different-dimensional space. We also develop a novel decomposition of the integral probability metric for the error analysis of bidirectional GANs. These basic theoretical results are of independent interest and can be applied to other related learning problems.

---
*Corresponding authors

35th Conference on Neural Information Processing Systems (NeurIPS 2021).

# 1 Introduction

Generative adversarial networks (GAN) (Goodfellow et al., 2014) is an important approach to implicitly learning and sampling from high-dimensional complex distributions. GANs have been shown to achieve impressive performance in many machine learning tasks (Radford et al., 2016; Reed et al., 2016; Zhu et al., 2017; Karras et al., 2018, 2019; Brock et al., 2019). Several recent studies have generalized GANs to bidirectional generative learning, which learns an encoder mapping the data distribution to the reference distribution simultaneously together with the generator doing reversely. These studies include the adversarial autoencoder (AAE) (Makhzani et al., 2015), bidirectional GAN (BiGAN) (Donahue et al., 2016), adversarially learned inference (ALI) (Dumoulin et al., 2016), and bidirectional generative modeling using adversarial gradient estimation (AGES) (Shen et al., 2020). A common feature of these methods is that they generalize the basic adversarial training framework of the original GAN from unidirectional to bidirectional. Dumoulin et al. (2016) showed that BiGANs make use of the joint distribution of data and latent representations, which can better capture the information of data than the vanilla GANs. Comparing with the unidirectional GANs, the joint distribution matching in the training of bidirectional GANs alleviates mode dropping and encourages cycle consistency (Shen et al., 2020).

Several elegant and stimulating papers have analyzed the theoretical properties of unidirectional GANs. Arora et al. (2017) considered the generalization error of GANs under the neural net distance. Zhang et al. (2018) improved the generalization error bound in Arora et al. (2017). Liang (2020) studied the minimax optimal rates for learning distributions with empirical samples under Sobolev evaluation class and density class. The minimax rate is $O(n^{-1/2} \vee n^{-\alpha+\beta/(2\alpha+\beta)})$, where $\alpha$ and $\beta$ are the regularity parameters for Sobolev density and evaluation class, respectively. Bai et al. (2019) analyzed the estimation error of GANs under the Wasserstein distance for a special class of distributions implemented by a generator, while the discriminator is designed to guarantee zero bias. Chen et al. (2020) studies the convergence properties of GANs when both the evaluation class and the target density class are Hölder classes and derived $O(n^{-\beta/(2\beta+d)} \log^2 n)$ bound, where $d$ is the dimension of the data distribution and $\alpha$ and $\beta$ are the regularity parameters for Hölder density and evaluation class, respectively. While impressive progresses have been made on the theoretical understanding of GANs, there are still some drawbacks in the existing results. For example,

(a) The reference distribution and the target data distribution are assumed to have the same dimension, which is not the actual setting for GAN training.
(b) The reference and the target data distributions are assumed to be supported on bounded sets.
(c) The prefactors in the convergence rates may depend on the dimension $d$ of the data distribution exponentially.

In practice, GANs are usually trained using a reference distributions with a lower dimension than that of the target data distribution. Indeed, an important strength of GANs is that they can model low-dimensional latent structures via using a low-dimensional reference distribution. The bounded support assumption excludes some commonly used Gaussian distributions as the reference. Therefore, strictly speaking, the existing convergence analysis results do not apply to what have been done in practice. In addition, there has been no theoretical analysis of bidirectional GANs in the literature.

## 1.1 Contributions

We derive nearly sharp non-asymptotic bounds for the GAN estimation error under the Dudley distance between the reference joint distribution and the data joint distribution. To the best of our knowledge, this is the first result providing theoretical guarantees for bidirectional GAN estimation error rate. We do not assume that the reference and the target data distributions have the same dimension or these distributions have bounded support. Also, our results are applicable to the Wasserstein distance if the target data distribution is assumed to have a bounded support.

The main novel aspects of our work are as follows.

(1) We allow the dimension of the reference distribution to be different from the dimension of the target distribution, in particular, it can be much lower than that of the target distribution.
(2) We allow unbounded support for the reference distribution and the target distribution under mild conditions on the tail probabilities of the target distribution.

(3) We explicitly establish that the prefactors in the error bounds depend on the square root of the dimension of the target distribution. This is a significant improvement over the exponential dependence on $d$ in the existing works.

Moreover, we develop a novel decomposition of integral probability metric for the error analysis of bidirectional GANs. We also show that the pushforward distribution of an empirical distribution based on neural networks can perfectly approximate another arbitrary empirical distribution as long as the number of discrete points are the same.

**Notation** We use $\sigma$ to denote the ReLU activation function in neural networks, which is $\sigma(x) = \max\{x, 0\}, x \in \mathbb{R}$. We use $I$ to denote the identity map. Without further indication, $\|\cdot\|$ represents the $L_2$ norm. For any function $g$, let $\|g\|_\infty = \sup_x \|g(x)\|$. We use notation $O(\cdot)$ and $\tilde{O}(\cdot)$ to express the order of function slightly differently, where $O(\cdot)$ omits the universal constant independent of $d$ while $\tilde{O}(\cdot)$ omits the constant depending on $d$. We use $B_2^d(a)$ to denote the $L_2$ ball in $\mathbb{R}^d$ with center at $\mathbf{0}$ and radius $a$. Let $g_{\#}\nu$ be the pushforward distribution of $\nu$ by function $g$ in the sense that $g_{\#}\nu(A) = \nu(g^{-1}(A))$ for any measurable set $A$. We use $\hat{\mathbb{E}}$ to denote taking expectation with respect to the empirical distribution.

## 2 Bidirectional generative learning

We describe the setup of the bidirectional GAN estimation problem and present the assumptions we need in our analysis.

### 2.1 Bidirectional GAN estimators

Let $\mu$ be the target data distribution supported on $\mathbb{R}^d$ for $d \geq 1$. Let $\nu$ be a reference distribution which is easy to sample from. We first consider the case when $\nu$ is supported on $\mathbb{R}$, and then extend it to $\mathbb{R}^k$, where $k \geq 1$ can be different from $d$. Usually, $k \ll d$ in practical machine learning tasks such as image generation. The goal is to learn functions $g : \mathbb{R} \to \mathbb{R}^d$ and $e : \mathbb{R}^d \to \mathbb{R}$ such that $\tilde{g}_{\#}\nu = \tilde{e}_{\#}\mu$, where $\tilde{g} := (g, I)$ and $\tilde{e} := (I, e)$, $\tilde{g}_{\#}\nu$ is the pushforward distribution of $\nu$ under $\tilde{g}$ and $\tilde{e}_{\#}\mu$ is the pushforward distribution of $\mu$ under $\tilde{e}$. We call $\tilde{g}_{\#}\nu$ the joint latent distribution or joint reference distribution and $\tilde{e}_{\#}\mu$ the joint data distribution or joint target distribution. At the population level, the bidirectional GAN solves the minimax problem:

$$(g^*, e^*, f^*) \in \arg \min_{g \in \mathcal{G}, e \in \mathcal{E}} \max_{f \in \mathcal{F}} \mathbb{E}_{Z \sim \nu}[f(g(Z), Z] - \mathbb{E}_{x \sim \mu}[f(X, e(X))],$$

where $\mathcal{G}, \mathcal{E}, \mathcal{F}$ are referred to as the generator class, the encoder class, and the discriminator class, respectively. Suppose we have two independent random samples $Z_1, \ldots, Z_n \overset{i.i.d.}{\sim} \nu$ and $X_1, \ldots, X_n \overset{i.i.d.}{\sim} \mu$. At the sample level, the bidirectional GAN solves the empirical version of the above minimax problem:

$$(\hat{g}_\theta, \hat{e}_\varphi, \hat{f}_\omega) = \arg \min_{g_\theta \in \mathcal{G}_{NN}, e_\varphi \in \mathcal{E}_{NN}} \max_{f_\omega \in \mathcal{F}_{NN}} \frac{1}{n} \sum_{i=1}^n f_\omega(g_\theta(Z_i), Z_i) - \frac{1}{n} \sum_{j=1}^n f_\omega(X_j, e_\varphi(X_j)),$$

$$(2.1)$$

where $\mathcal{G}_{NN}$ and $\mathcal{E}_{NN}$ are two classes of neural networks approximating the generator class $\mathcal{G}$ and the encoder class $\mathcal{E}$ respectively, and $\mathcal{F}_{NN}$ is a class of neural networks approximating the discriminator class $\mathcal{F}$.

### 2.2 Assumptions

We assume the target $\mu$ and the reference $\nu$ satisfy the following assumptions.

**Assumption 1** (Subexponential tail). *For a large $n$, the target distribution $\mu$ on $\mathbb{R}^d$ and the reference distribtuion $\nu$ on $\mathbb{R}$ satisfies the first moment tail condition for some $\delta > 0$,*

$$\max\{\mathbb{E}_\nu \|Z\| \mathbb{1}_{\{\|Z\| > \log n\}}, \mathbb{E}_\mu \|X\| \mathbb{1}_{\{\|X\| > \log n\}}\} = O(n^{-\frac{(\log n)^\delta}{d}}).$$

**Assumption 2** (Absolute continuity). *Both the target distribution $\mu$ on $\mathbb{R}^d$ and the reference distribution $\nu$ on $\mathbb{R}$ are absolutely continuous with respect to the Lebesgue measure $\lambda$.*

Assumption 1 is a technical condition for dealing with the case when $\mu$ and $\nu$ are supported on $\mathbb{R}^d$ and $\mathbb{R}$ instead of compact subsets. For distributions with bounded supports, this assumption is automatically satisfied. Here the factor $(\log n)^\delta$ ensures that the tails of $\mu$ and $\nu$ are sub-exponential, and it can be easily satisfied if the distributions are sub-gaussian. For the reference distribution, Assumption 1 and 2 can be easily satisfied by specifying $\nu$ as some common distribution with easy-to-sample density such as Gaussian or uniform, which is usually done in the applications of GANs. For the target distribution, Assumption 1 and 2 specifies the type of distributions that are learnable by bidirectional GAN with our theoretical guarantees. Note that Assumption 1 is also necessary in our proof for bounding the generator and encoder approximation error in the sense that the results will not hold if we replace $(\log n)^\delta$ with 1. Assumption 2 is also necessary for Theorem 4.3 in mapping between empirical samples, which is essential in bounding generator and encoder approximation error.

## 2.3 Generator, encoder and discriminator classes

Let $\mathcal{F}_{NN} := \mathcal{NN}(W_1, L_1)$ be the discriminator class consisting of the feedforward ReLU neural networks $f_\omega : \mathbb{R}^{d+1} \mapsto \mathbb{R}$ with width $W_1$ and depth $L_1$. Similarly, let $\mathcal{G}_{NN} := \mathcal{NN}(W_2, L_2)$ be the generator class consisting of the feedforward ReLU neural networks $g_\theta : \mathbb{R} \mapsto \mathbb{R}^d$ with width $W_2$ and depth $L_2$, and $\mathcal{E}_{NN} := \mathcal{NN}(W_3, L_3)$ the encoder class consisting of the feedforward ReLU neural networks $e_\varphi : \mathbb{R}^d \mapsto \mathbb{R}$ with width $W_3$ and depth $L_3$.

The functions $f_\omega \in \mathcal{F}_{NN}$ have the following form:

$$f_\omega(x) = A_{L_1} \cdot \sigma(A_{L_1-1} \cdots \sigma(A_1 x + b_1) \cdots + b_{L_1-1}) + b_{L_1}$$

where $A_i$ are the weight matrices with number of rows and columns no larger than the width $W_1$, $b_i$ are the bias vectors with compatible dimensions, and $\sigma$ is the ReLU activation function $\sigma(x) = x \vee 0$. Similarly, functions $g_\theta \in \mathcal{G}_{NN}$ and $e_\varphi \in \mathcal{E}_{NN}$ have the following form:

$$g_\theta(x) = A'_{L_2} \cdot \sigma(A'_{L_2-1} \cdots \sigma(A'_1 x + b'_1) \cdots + b'_{L_2-1}) + b'_{L_2}$$
$$e_\varphi(x) = A''_{L_3} \cdot \sigma(A''_{L_3-1} \cdots \sigma(A''_1 x + b''_1) \cdots + b''_{L_3-1}) + b''_{L_3}$$

where $A'_i$ and $A''_i$ are the weight matrices with number of rows and columns no larger than $W_2$ and $W_3$, respectively, and $b'_i$ and $b''_i$ are the bias vectors with compatible dimensions.

We impose the following conditions on $\mathcal{G}_{NN}$, $\mathcal{E}_{NN}$, and $\mathcal{F}_{NN}$.

**Condition 1.** *For any $g_\theta \in \mathcal{G}_{NN}$ and $e_\varphi \in \mathcal{E}_{NN}$, we have $\max\{\|g_\theta\|_\infty, \|e_\varphi\|_\infty\} \le \log n$.*

Condition 1 on $\mathcal{G}_{NN}$ can be easily satisfied by adding an additional clipping layer $\ell$ after the original output layer, with $c_{n,d} \equiv (\log n)/\sqrt{d}$,

$$\ell(a) = a \wedge c_{n,d} \vee (-c_{n,d}) = \sigma(a + c_{n,d}) - \sigma(a - c_{n,d}) - c_{n,d}. \tag{2.2}$$

We truncate the output of $\|g_\theta\|$ to an increasing interval $[-\log n, \log n]$ to include the whole $\mathbb{R}^d$ support for the evaluation function class. Condition 1 on $\mathcal{E}_{NN}$ can be satisfied in the same manner. This condition is technically necessary in our proof (see appendix).

# 3 Non-asymptotic error bounds

We characterize the bidirectional GAN solutions based on minimizing the integral probability metric (IPM, Müller (1997)) between two distributions $\mu$ and $\nu$ with respect to a symmetric evaluation function class $\mathcal{F}$, defined by

$$d_\mathcal{F}(\mu, \nu) = \sup_{f \in \mathcal{F}} [\mathbb{E}_\mu f - \mathbb{E}_\nu f]. \tag{3.1}$$

By specifying the evaluation function class $\mathcal{F}$ differently, we can obtain many commonly-used metrics (Liu et al., 2017). Here we focus on the following two

- $\mathcal{F} = $ bounded Lipschitz function class : $d_\mathcal{F} = d_{BL}$, (bounded Lipschitz (or Dudley) metric: metrizing weak convergence, Dudley (2018)),
- $\mathcal{F} = $ Lipschitz function class : $d_\mathcal{F} = W_1$ (Wasserstein GAN, Arjovsky et al. (2017)).

We consider the estimation error under the Dudley metric $d_{BL}$. Note that in the case when $\mu$ and $\nu$ have bounded supports, the Dudley metric $d_{BL}$ is equivalent to the 1-Wasserstein metric $W_1$. Therefore, under the bounded support condition for $\mu$ and $\nu$, all our convergence results also hold under the Wasserstein distance $W_1$. Even if the support of $\mu$ and $\nu$ are unbounded, we can still apply the result of Lu and Lu (2020) to avoid empirical process theory and obtain an stochastic error bound under the Wasserstein distance $W_1$. However, the result of Lu and Lu (2020) requires sub-gaussianity to obtain the $\sqrt{d}$ prefactor. In order to make it more general, we use the empirical processes theory to get the explicit prefactor. Also, the discriminator approximation error will be unbounded if we consider the Wasserstein distance $W_1$. Hence, we can only consider $d_{BL}$ for the unbounded support case.

The bidirectional GAN solution $(\hat{g}_\theta, \hat{e}_\varphi)$ in (2.1) also minimizes the distance between $(\tilde{g}_\theta)_{\#}\hat{\nu}_n$ and $(\tilde{e}_\varphi)_{\#}\hat{\mu}_n$ under $d_{\mathcal{F}_{NN}}$

$$\min_{g_\theta \in \mathcal{G}_{NN}, e_\varphi \in \mathcal{E}_{NN}} d_{\mathcal{F}_{NN}}((\tilde{g}_\theta)_{\#}\hat{\nu}_n, (\tilde{e}_\varphi)_{\#}\hat{\mu}_n).$$

However, even if two distributions are close with respect to $d_{\mathcal{F}_{NN}}$, there is no automatic guarantee that they will still be close under other metrics, for example, the Dudley or the Wasserstein distance (Arora et al., 2017). Therefore, it is natural to ask the question:

- How close are the two bidirectional GAN estimators $\hat{\boldsymbol{\nu}} := (\hat{g}_\theta, I)_{\#}\nu$ and $\hat{\boldsymbol{\mu}} := (I, \hat{e}_\varphi)_{\#}\mu$ under some other stronger metrics?

We consider the IPM with the uniformly bounded 1-Lipschitz function class on $\mathbb{R}^{d+1}$, as the evaluation class, which is defined as, for some finite $B > 0$,

$$\mathcal{F}^1 := \left\{ f : \mathbb{R}^{d+1} \mapsto \mathbb{R} \,\middle|\, |f(x) - f(y)| \le \|x - y\|, x, y \in \mathbb{R}^{d+1} \text{ and } \|f\|_\infty \le B \right\} \qquad (3.2)$$

In Theorem 3.1, we consider the bounded support case where $d_{\mathcal{F}} = W_1$; In Theorem 3.2, we extend the result to the unbounded support case; In Theorem 3.3, we extend the result to the case where the dimension of the reference distribution is arbitrary.

We first present a result when $\mu$ is supported on a compact subset $[-M, M]^d \subset \mathbb{R}^d$ and $\nu$ is supported on $[-M, M] \subset \mathbb{R}$ for a finite $M > 0$.

**Theorem 3.1.** *Suppose that the target $\mu$ is supported on $[-M, M]^d \subset \mathbb{R}^d$ and the reference $\nu$ is supported on $[-M, M] \subset \mathbb{R}$ for a finite $M > 0$, and Assumption 2 holds. Let the outputs of $g_\theta$ and $e_\varphi$ be within $[-M, M]^d$ and $[-M, M]$ for $g_\theta \in \mathcal{G}_{NN}$ and $e_\varphi \in \mathcal{E}_{NN}$, respectively. By specifying the three network structures as $W_1 L_1 \ge \lceil \sqrt{n} \rceil$, $W_2^2 L_2 = C_1 dn$, and $W_3^2 L_3 = C_2 n$ for some constants $12 \le C_1, C_2 \le 384$ and properly choosing parameters, we have*

$$\mathbb{E} d_{\mathcal{F}^1}(\hat{\boldsymbol{\nu}}, \hat{\boldsymbol{\mu}}) \le C_0 \sqrt{d} n^{-\frac{1}{d+1}} (\log n)^{\frac{1}{d+1}},$$

*where $C_0 > 0$ is a constant independent of $d$ and $n$.*

The prefactor $C_0 \sqrt{d}$ in the error bound depends on $d^{1/2}$ linearly. This is different from the existing works where the dependence of the prefactor on $d$ is either not clearly described or is exponential. In high-dimensional settings with large $d$, this makes a substantial difference in the quality of the error bounds. These remarks apply to all the results stated below.

The next theorem deals with the case of unbounded support.

**Theorem 3.2.** *Suppose Assumption 1 and 2 hold, and Condition 1 is satisfied. By specifying the structures of the three network classes as $W_1 L_1 \ge \lceil \sqrt{n} \rceil$, $W_2^2 L_2 = C_1 dn$, and $W_3^2 L_3 = C_2 n$ for some constants $12 \le C_1, C_2 \le 384$ and properly choosing parameters, we have*

$$\mathbb{E} d_{\mathcal{F}^1}(\hat{\boldsymbol{\nu}}, \hat{\boldsymbol{\mu}}) \le \min \left\{ C_0 \sqrt{d} n^{-\frac{1}{d+1}} (\log n)^{1 + \frac{1}{d+1}}, C_d n^{-\frac{1}{d+1}} \log n \right\},$$

*where $C_0$ is a constant independent of $d$ and $n$, but $C_d$ depends on $d$.*

Note that two methods are used in bounding stochastic errors (see appendix), which leads to two different bounds: one with an explicit $\sqrt{d}$ prefactor with the cost that we have an additional $\log n$ factor. Another one with an implicit prefactor but with a better $\log n$ factor. Hence, it is a tradeoff between the explicitness of prefactor and the order of $\log n$.

Our next result generalizes the results to the case when the reference distribution $\nu$ is supported on $\mathbb{R}^k$ for $k \in \mathbb{N}_+$.

**Assumption 3.** *The target distribution $\mu$ on $\mathbb{R}^d$ is absolutely continuous with respect to the Lebesgue measure on $\mathbb{R}^d$ and the reference distribution $\nu$ on $\mathbb{R}^k$ is absolutely continuous with respect to the Lebesgue measure on $\mathbb{R}^k$, and $k \ll d$.*

With the above assumption, we have the following theorem providing theoretical guarantees for the validity of any dimensional reference $\nu$.

**Theorem 3.3.** *Suppose Assumption 1 and 3 hold, and Condition 1 is satisfied. By specifying generator and discriminator class structure as $W_1 L_1 \geq \lceil \sqrt{n} \rceil$, $W_2^2 L_2 = C_1 dn$, and $W_3^2 L_3 = C_2 kn$ for some constants $12 \leq C_1, C_2 \leq 384$ and properly choosing parameters, we have*

$$\mathbb{E}d_{\mathcal{F}^1}(\hat{\boldsymbol{\nu}}, \hat{\boldsymbol{\mu}}) \leq \min\left\{ C_0 \sqrt{d} n^{-\frac{1}{d+k}} (\log n)^{1+\frac{1}{d+k}}, C_d n^{-\frac{1}{d+k}} \log n \right\},$$

*where $C_0$ is a constant independent of $d$ and $n$, but $C_d$ depends on $d$.*

Note that the errors bounds established in Theorems 3.1-3.3 are tight up to a logarithmic factor, since the minimax rate measured in Wasserstein distance for learning distributions when the Lipschitz evaluation class is defined on $\mathbb{R}^d$ is $\tilde{O}(n^{-\frac{1}{d}})$ (Liang, 2020).

# 4 Approximation and stochastic errors

In this section we present a novel inequality for decomposing the total error into approximation and stochastic errors and establish bounds on these errors.

## 4.1 Decomposition of the estimation error

Define the approximation error of a function class $\mathcal{F}$ to another function class $\mathcal{H}$ by

$$\mathcal{E}(\mathcal{H}, \mathcal{F}) := \sup_{h \in \mathcal{H}} \inf_{f \in \mathcal{F}} \|h - f\|_\infty.$$

We decompose the Dudley distance $d_{\mathcal{F}^1}(\hat{\boldsymbol{\nu}}, \hat{\boldsymbol{\mu}})$ between the latent joint distribution and the data joint distribution into four different error terms,

- the approximation error of the discriminator class $\mathcal{F}_{NN}$ to $\mathcal{F}^1$:

$$\mathcal{E}_1 = \mathcal{E}(\mathcal{F}^1, \mathcal{F}_{NN}),$$

- the approximation error of the generator and encoder classes:

$$\mathcal{E}_2 = \inf_{g_\theta \in \mathcal{G}_{NN}, e_\varphi \in \mathcal{E}_{NN}} \sup_{f_\omega \in \mathcal{F}_{NN}} \frac{1}{n} \sum_{i=1}^n \left( f_\omega(g_\theta(z_i), z_i) - f_\omega(x_i, e_\varphi(x_i)) \right),$$

- the stochastic error for the latent joint distribution $\hat{\boldsymbol{\nu}}$:

$$\mathcal{E}_3 = \sup_{f_\omega \in \mathcal{F}^1} \mathbb{E}f_\omega(\hat{g}(z), z) - \hat{\mathbb{E}}f_\omega(\hat{g}(z), z),$$

- the stochastic error for the latent joint distribution $\hat{\boldsymbol{\mu}}$:

$$\mathcal{E}_4 = \sup_{f_\omega \in \mathcal{F}^1} \hat{\mathbb{E}}f_\omega(x, \hat{e}(x)) - \mathbb{E}f_\omega(x, \hat{e}(x)).$$

**Lemma 4.1.** *Let $(\hat{g}_\theta, \hat{e}_\varphi)$ be the bidirectional GAN solution in (2.1) and $\mathcal{F}^1$ be the uniformly bounded 1-Lipschitz function class defined in (3.2). Then the Dudley distance between the latent joint distribution $\hat{\boldsymbol{\nu}} = (\hat{g}_\theta, I)_{\#}\nu$ and the data joint distribution $\hat{\boldsymbol{\mu}} = (I, \hat{e}_\varphi)_{\#}\mu$ can be decomposed as follows*

$$d_{\mathcal{F}^1}(\hat{\boldsymbol{\nu}}, \hat{\boldsymbol{\mu}}) \quad \leq \quad 2\mathcal{E}_1 + \mathcal{E}_2 + \mathcal{E}_3 + \mathcal{E}_4. \tag{4.1}$$

The novel decomposition (4.1) is fundamental to our error analysis. Based on (4.1), we bound each error term on the right side of (4.1) and balance the bounds to obtain an overall bound for the bidirectional GAN estimation.

For proving Lemma 4.1, we introduce the following useful inequality, which states that for any two probability distributions, the difference in IPMs with two distinct evaluation classes will not exceed 2 times the approximation error between the two evaluation classes, that is, for any probability distributions $\mu$ and $\nu$ and symmetric function classes $\mathcal{F}$ and $\mathcal{H}$,

$$d_{\mathcal{H}}(\mu, \nu) - d_{\mathcal{F}}(\mu, \nu) \leq 2\mathcal{E}(\mathcal{H}, \mathcal{F}). \tag{4.2}$$

It is easy to check that if we replace $d_{\mathcal{H}}(\mu, \nu)$ by $\hat{d}_{\mathcal{H}}(\mu, \nu) := \sup_{h \in \mathcal{H}} [\hat{\mathbb{E}}_{\mu} h - \hat{\mathbb{E}}_{\nu} h]$, (4.2) still holds.

*Proof of Lemma 4.1.* We have

$$
\begin{aligned}
d_{\mathcal{F}^1}(\hat{\boldsymbol{\nu}}, \hat{\boldsymbol{\mu}}) &= \sup_{f_\omega \in \mathcal{F}^1} \mathbb{E} f_\omega(\hat{g}(z), z) - \mathbb{E} f_\omega(x, \hat{e}(x)) \\
&\leq \sup_{f_\omega \in \mathcal{F}^1} \mathbb{E} f_\omega(\hat{g}(z), z) - \hat{\mathbb{E}} f_\omega(\hat{g}(z), z) + \sup_{f_\omega \in \mathcal{F}^1} \hat{\mathbb{E}} f_\omega(\hat{g}(z), z) - \hat{\mathbb{E}} f_\omega(x, \hat{e}(x)) \\
&\quad + \sup_{f_\omega \in \mathcal{F}^1} \hat{\mathbb{E}} f_\omega(x, \hat{e}(x)) - \mathbb{E} f_\omega(x, \hat{e}(x)) \\
&= \mathcal{E}_3 + \mathcal{E}_4 + \sup_{f_\omega \in \mathcal{F}^1} \hat{\mathbb{E}} f_\omega(\hat{g}(z), z) - \hat{\mathbb{E}} f_\omega(x, \hat{e}(x)).
\end{aligned}
$$

Denote $A := \sup_{f_\omega \in \mathcal{F}^1} \hat{\mathbb{E}} f_\omega(\hat{g}(z), z) - \hat{\mathbb{E}} f_\omega(x, \hat{e}(x))$. By (4.2) and the optimality of the bidirectional GAN solutions, $A$ satisfies

$$
\begin{aligned}
A &= \sup_{f_\omega \in \mathcal{F}^1} \frac{1}{n} \sum_{i=1}^{n} \Big( f_\omega(\hat{g}(z_i), z_i) - f_\omega(x_i, \hat{e}(x_i)) \Big) \\
&\leq \sup_{f_\omega \in \mathcal{F}_{NN}} \frac{1}{n} \sum_{i=1}^{n} \Big( f_\omega(\hat{g}(z_i), z_i) - f_\omega(x_i, \hat{e}(x_i)) \Big) + 2\mathcal{E}(\mathcal{F}^1, \mathcal{F}_{NN}) \\
&= \inf_{g_\theta \in \mathcal{G}_{NN}, e_\varphi \in \mathcal{E}_{NN}} \sup_{f_\omega \in \mathcal{F}_{NN}} \frac{1}{n} \sum_{i=1}^{n} \Big( f_\omega(g_\theta(z_i), z_i) - f_\omega(x_i, e_\varphi(x_i)) \Big) + 2\mathcal{E}_1 \\
&= 2\mathcal{E}_1 + \mathcal{E}_2.
\end{aligned}
$$

$\square$

Note that we cannot directly apply the symmetrization technic (see appendix) to $\mathcal{E}_3$ and $\mathcal{E}_4$ since $e^*$ and $g^*$ are correlated with $x_i$ and $z_i$. However, this problem can be solved by replacing the samples $(x_i, z_i)$ in the empirical terms in $\mathcal{E}_3$ and $\mathcal{E}_4$ with ghost samples $(x'_i, z'_i)$ independent of $(x_i, z_i)$ and replacing $g^*$ and $e^*$ with $g^{**}$ and $e^{**}$ which are obtained from the ghost samples, respectively. That is, we replace $\hat{\mathbb{E}} f_\omega(g^*(z), z)$ and $\hat{\mathbb{E}} f_\omega(x, e^*(x))$ with $\hat{\mathbb{E}} f_\omega(g^{**}(z'), z')$ and $\hat{\mathbb{E}} f_\omega(x', e^{**}(x'))$ in $\mathcal{E}_3$ and $\mathcal{E}_4$, respectively. Then we can proceed with the same proof of Lemma 4.1 and apply the symmetrization technic to $\mathcal{E}_3$ and $\mathcal{E}_4$, since $(g^*(z_i), z_i)$ and $(g^{**}(z'_i), z'_i)$ have the same distribution. To simplify the notation, we will just use $\hat{\mathbb{E}} f_\omega(g^*(z), z)$ and $\hat{\mathbb{E}} f_\omega(x, e^*(x))$ to denote $\hat{\mathbb{E}} f_\omega(g^{**}(z'), z')$ and $\hat{\mathbb{E}} f_\omega(x', e^{**}(x'))$ here, respectively.

## 4.2 Approximation errors

We now discuss the errors due to the discriminator approximation and the generator and encoder approximation.

### 4.2.1 The discriminator approximation error $\mathcal{E}_1$

The discriminator approximation error $\mathcal{E}_1$ describes how well the discriminator neural network class approximates functions from the Lipschitz class $\mathcal{F}^1$. Lemma 4.2 below can be applied to obtain the neural network approximation error for Lipschitz functions. It leads to a quantitative and non-asymptotic approximation rate in terms of the width and depth of the neural networks when bounding $\mathcal{E}_1$.

**Lemma 4.2** (Shen et al. (2021)). *Let $f$ be a Lipschitz continuous function defined on $[-R, R]^d$. For arbitrary $W, L \in \mathbb{N}_+$, there exists a function $\psi$ implemented by a ReLU feedforward neural network with width $W$ and depth $L$ such that*

$$\|f - \psi\|_\infty = O\big(\sqrt{d}R(WL)^{-\frac{2}{d}}\big).$$

By Lemma 4.2 and our choice of the architecture of discriminator class $\mathcal{F}_{NN}$ in the theorems, we have $\mathcal{E}_1 = O\big(\sqrt{d}(W_1 L_1)^{-\frac{2}{d+1}} \log n\big)$. Theorem 4.2 also informs about how to choose the architecture of the discriminator networks based on how small we want the approximation error $\mathcal{E}_1$ to be. By setting $(W_1 L_1)^2 \geq n$, $\mathcal{E}_1$ is dominated by the stochastic terms $\mathcal{E}_3$ and $\mathcal{E}_4$.

#### 4.2.2 The generator and encoder approximation error $\mathcal{E}_2$

The generator and encoder approximation error $\mathcal{E}_2$ describes how powerful the generator and encoder classes are in pushing the empirical distributions $\hat{\mu}_n$ and $\hat{\nu}_n$ to each other. A natural question is

- Can we find some generator and encoder neural network functions such that $\mathcal{E}_2 = 0$?

Most of the current literature concerning the error analysis of GANs applied the optimal transport theory (Villani, 2008) to minimize an error term similar to $\mathcal{E}_2$, see, for example, Chen et al. (2020). However, the existence of the optimal transport function from $\mathbb{R} \to \mathbb{R}^d$ is not guaranteed. Therefore, the existing analysis of GANs can only deal with the scenario when the reference and the target data distribution are assumed to have the same dimension. This equal dimensionality assumption is not satisfied in the actual training of GANs or bidirectional GANs in many applications. Here, instead of using the optimal transport theory, we establish the following approximation results in Theorem 4.3, which enables us to forgo the equal dimensionality assumption.

**Theorem 4.3.** *Suppose that $\nu$ supported on $\mathbb{R}$ and $\mu$ supported on $\mathbb{R}^d$ are both absolutely continuous w.r.t. the Lebesgue measures, and $z_i's$ and $x_i's$ are i.i.d. samples from $\nu$ and $\mu$, respectively for $1 \leq i \leq n$. Then there exist generator and encoder neural network functions $g : \mathbb{R} \mapsto \mathbb{R}^d$ and $e : \mathbb{R}^d \mapsto \mathbb{R}$ such that $g$ and $e$ are inverse bijections of each other between $\{z_i : 1 \leq i \leq n\}$ and $\{x_i : 1 \leq i \leq n\}$ up to a permutation. Moreover, such neural network functions $g$ and $e$ can be obtained by properly specifying $W_2^2 L_2 = c_2 dn$ and $W_3^2 L_3 = c_3 n$ for some constant $12 \leq c_2, c_3 \leq 384$.*

*Proof.* By the absolute continuity of $\nu$ and $\mu$, all the $z_i's$ and $x_i's$ are distinct a.s.. We can reorder $z_i's$ from the smallest to the largest, so $z_1 < z_2 < \ldots < z_n$. Let $z_{i+1/2}$ be any point between $z_i$ and $z_{i+1}$ for $i \in \{1, 2, \ldots, n-1\}$. We define the continuous piece-wise linear function $g : \mathbb{R} \mapsto \mathbb{R}^d$ by

$$g(z) = \begin{cases} x_1 & z \leq z_1, \\ \frac{z - z_{i+1/2}}{z_i - z_{i+1/2}} x_i + \frac{z - z_i}{z_{i+\frac{1}{2}} - z_i} x_{i+1} & z = (z_i, z_{i+1/2}), \text{ for } i = 1, \ldots, n-1, \\ x_{i+1} & z \in [z_{i+1/2}, z_{i+1}], \text{ for } i = 1, \ldots, n-2, \\ x_n & z \geq z_{n-1+1/2}. \end{cases}$$

By Yang et al. (2021, Lemma 3.1), $g \in \mathcal{NN}(W_2, L_2)$ if $n \leq (W_2 - d - 1) \lfloor \frac{W_2 - d - 1}{6d} \rfloor \lfloor \frac{L_2}{2} \rfloor$. Taking $n = (W_2 - d - 1) \lfloor \frac{W_2 - d - 1}{6d} \rfloor \lfloor \frac{L_2}{2} \rfloor$, a simple calculation shows $W_2^2 L_2 = cdn$ for some constant $12 \leq c \leq 384$. The existence of neural net function $e$ can be constructed in the same way due to the fact that the first coordinate of $x_i's$ are distinct almost surely. $\qquad \square$

When the number of point masses of the empirical distributions are relatively moderate compared with the structure of the neural nets, we can approximate empirical distributions arbitrarily well with any empirical distribution with the same number of point masses pushforwarded by the neural nets.

Theorem 4.3 provides an effective way to specify the architecture of generator and encoder classes. According to this lemma, we can take $n = \frac{W_2 - d}{2} \lfloor \frac{W_2 - d}{6d} \rfloor \lfloor \frac{L_2}{2} \rfloor + 2 = \frac{W_3 - 1}{2} \lfloor \frac{W_3 - 1}{6} \rfloor \lfloor \frac{L_3}{2} \rfloor + 2$, which gives rise to $W_2^2 L_2 / d \asymp W_3^2 L_3 \asymp n$. More importantly, Theorem 4.3 can be applied to

bound $\mathcal{E}_2$ as follows.

$$\mathcal{E}_2 = \inf_{g_\theta \in \mathcal{G}_{NN}, e_\varphi \in \mathcal{E}_{NN}} \sup_{f_\omega \in \mathcal{F}_{NN}} \frac{1}{n} \sum_{i=1}^n \Big( f_\omega(g_\theta(z_i), z_i) - f_\omega(x_i, e_\varphi(x_i)) \Big)$$

$$\leq \inf_{g_\theta \in \mathcal{G}_{NN}} \sup_{f_\omega \in \mathcal{F}_{NN}} \frac{1}{n} \sum_{i=1}^n \Big( f_\omega(g_\theta(z_i), z_i) - f_\omega(x_i, z_i) \Big)$$

$$+ \inf_{e_\varphi \in \mathcal{E}_{NN}} \sup_{f_\omega \in \mathcal{F}_{NN}} \frac{1}{n} \sum_{i=1}^n \Big( f_\omega(x_i, z_i) - f_\omega(x_i, e_\varphi(x_i)) \Big)$$

$$= 0.$$

We simply reordered $z_i's$ and $x_i's$ as in the proof. Therefore, this error term can be perfectly eliminated.

### 4.3 Stochastic errors

The stochastic error $\mathcal{E}_3$ ($\mathcal{E}_4$) quantifies how close the empirical distribution and the true latent joint distribution (data joint distribution) are with the Lipschitz class $\mathcal{F}^1$ as the evaluation class under IPM. We apply the results in the refined Dudley inequality (Schreuder, 2020) in Lemma 4.4 to bound $\mathcal{E}_3$ and $\mathcal{E}_4$.

**Lemma 4.4** (Refined Dudley Inequality). *For a symmetric function class $\mathcal{F}$ with $\sup_{f \in \mathcal{F}} ||f||_\infty \leq M$, we have*

$$\mathbb{E}[d_\mathcal{F}(\hat{\mu}_n, \mu)] \leq \inf_{0 < \delta < M} \left( 4\delta + \frac{12}{\sqrt{n}} \int_\delta^M \sqrt{\log \mathcal{N}(\epsilon, \mathcal{F}, || \cdot ||_\infty)} d\epsilon \right).$$

The original Dudley inequality (Dudley, 1967; Van der Vaart and Wellner, 1996) suffers from the problem that if the covering number $\mathcal{N}(\epsilon, \mathcal{F}, || \cdot ||_\infty)$ increases too fast as $\epsilon$ goes to 0, then the upper bound will be infinity, which is totally meaningless. The improved Dudley inequality circumvents such a problem by only allowing $\epsilon$ to integrate from $\delta > 0$ as is shown in Lemma 4.4, which also indicates that $\mathbb{E}\mathcal{E}_3$ scales with the covering number $\mathcal{N}(\epsilon, \mathcal{F}^1, || \cdot ||_\infty)$.

By calculating the covering number of $\mathcal{F}^1$ and utilizing the refined Dudley inequality, we can obtain the upper bound

$$\max\{\mathbb{E}\mathcal{E}_3, \mathbb{E}\mathcal{E}_4\} = O\left( C_d n^{-\frac{1}{d+1}} \log n \wedge \sqrt{d} n^{-\frac{1}{d+1}} (\log n)^{1 + \frac{1}{d+1}} \right). \tag{4.3}$$

## 5 Related work

Recently, several impressive works have studied the challenging problem of the convergence properties of unidirectional GANs. Arora et al. (2017) noted that training of GANs may not have good generalization properties in the sense that even if training may appear successful but the trained distribution may be far from target distribution in standard metrics. On the other hand, Bai et al. (2019) showed that GANs can learn distributions in Wasserstein distance with polynomial sample complexity. Liang (2020) studied the rates of convergence of a class of GANs, including Wasserstein, Sobolev and MMD GANs. This work also established the nonparametric minimax optimal rate under the Sobolev IPM. The results of Bai et al. (2019) and Liang (2020) require invertible generator networks, meaning all the weight matrices need to be full-rank, and the activation function needs to be the invertible leaky ReLU activation. Chen et al. (2020) established an upper bound for the estimation error rate under Hölder evaluation and target density classes, where $\mathcal{H}^\beta$ is Hölder class with regularity $\beta$ and the density of the target $\mu$ is assumed to belong to $\mathcal{H}^\alpha$. They assumed that the reference distribution has the same dimension as the target distribution and applied the optimal transport theory to control the generator approximation error. However, how the prefactor depends in the error bounds on the dimension $d$ in the existing results (Liang, 2020; Chen et al., 2020) is either not clearly described or is exponential. In high-dimensional settings with large $d$, this makes a substantial difference in the quality of the error bounds.

Singh et al. (2019) studied minimax convergence rates of nonparametric density estimation under a class of adversarial losses and investigated how the choice of loss and the assumed smoothness of the underlying density together determine the minimax rate; they also discussed connections to learning generative models in a minimax statistical sense. Uppal et al. (2019) generates the idea of Sobolev IPM to Besov IPM, where both target density and the evaluation classes are Besov classes. They also showed how their results imply bounds on the statistical error of a GAN.

These results provide important insights in the understanding of GANs. However, as we mentioned earlier, some of the assumptions made in these results, including equal dimension between the reference and target distributions and bounded support of the distributions, are not satisfied in the training of GANs in practice. Our results avoid these assumptions. Moreover, the prefactors in our error bounds are clearly described as being dependent on the square root of the dimension $d$. Finally, the aforementioned results only dealt with unidirectional GANs. Our work is the first to address the convergence properties of bidirectional GANs.

# 6    Conclusion

This paper derives the error bounds for the bidirectional GANs under the Dudley distance between the latent joint distribution and the data joint distribution. The results are established without the two crucial conditions that are commonly assumed in the existing literature: equal dimensionality between the reference and the target distributions and bounded support for these distributions. Additionally, this work contributes to the neural network approximation theory by constructing neural network functions such that the pushforward distribution of an empirical distribution can perfectly approximate another arbitrary empirical distribution with a different dimension as long as their number of point masses are equal. A novel decomposition of integral probability metric is also developed for error analysis of bidirectional GANs, which can be useful in other generative learning problems.

A limitation of our results, as well as all the existing results on the convergence properties of GANs, is that they suffer from the curse of dimensionality, which cannot be circumvented by assuming sufficient smoothness assumptions. In many applications, high-dimensional complex data such as images, texts and natural languages, tend to be supported on approximate lower-dimensional manifolds. It is desirable to take into such structure in the theoretical analysis. An important extension of the present results is to show that bidirectional GANs can circumvent the curse of dimensionality if the target distribution is assumed to be supported on an approximate lower-dimensional manifold. This appears to be a technically challenging problem and will be pursued in our future work.

# Acknowledgements

The authors wish to thank the three anonymous reviewers for their insightful comments and constructive suggestions that helped improve the paper significantly.

The work of J. Huang is partially supported by the U.S. NSF grant DMS-1916199. The work of Y. Jiao is supported in part by the National Science Foundation of China under Grant 11871474 and by the research fund of KLATASDSMOE. The work of Y. Wang is supported in part by the Hong Kong Research Grant Council grants 16308518 and 16317416 and HK Innovation Technology Fund ITS/044/18FX, as well as Guangdong-Hong Kong-Macao Joint Laboratory for Data-Driven Fluid Mechanics and Engineering Applications.

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
