# Supplementary Material for "Non-Asymptotic Error Bounds for Bidirectional GANs"

**Shiao Liu**
Department of Statistics and Actuarial Science, University of Iowa
Iowa City, IA 52242, USA
shiao-liu@uiowa.edu

**Yunfei Yang** [*]
Department of Mathematics, The Hong Kong University of Science and Technology
Clear Water Bay, Hong Kong, China
yyangdc@connect.ust.hk

**Jian Huang** [*]
Department of Statistics and Actuarial Science, University of Iowa
Iowa City, IA 52242, USA
jian-huang@uiowa.edu

**Yuling Jiao** [*]
School of Mathematics and Statistics, Wuhan University
Wuhan, Hubei, China 430072
yulingjiaomath@whu.edu.cn

**Yang Wang**
Department of Mathematics, The Hong Kong University of Science and Technology
Clear Water Bay, Hong Kong, China
yangwang@ust.hk

In this supplementary material, we first prove Theorem 3.2, and then Theorems 3.1 and 3.3.

## A  Notations and Preliminaries

We use $\sigma$ to denote the ReLU activation function in neural networks, which is $\sigma(x) = \max\{x, 0\}$. Without further indication, $\|\cdot\|$ represents the $L_2$ norm. For any function $g$, let $\|g\|_\infty = \sup_x \|g(x)\|$. We use notation $O(\cdot)$ and $\tilde{O}(\cdot)$ to express the order of function slightly differently, where $O(\cdot)$ omits the universal constant not relying on $d$ while $\tilde{O}(\cdot)$ omits the constant related to $d$. We use $B_2^d(a)$ to denote $L_2$ ball in $\mathbb{R}^d$ with center at $\mathbf{0}$ and radius $a$. Let $g_\#\nu$ be the pushforward distribution of $\nu$ by function $g$ in the sense that $g_\#\nu(A) = \nu(g^{-1}(A))$ for any measurable set $A$.

The $r$-**covering number** of some class $\mathcal{F}$ w.r.t. norm $\|\cdot\|$ is the minimum number of $r$-$\|\cdot\|$ radius balls needed to cover $\mathcal{F}$, which we denote as $\mathcal{N}(r, \mathcal{F}, \|\cdot\|)$. We denote $\mathcal{N}(r, \mathcal{F}, L_2(P_n))$ as the covering number of $\mathcal{F}$ w.r.t. $L_2(P_n)$, which is defined as $\|f\|_{L_2(P_n)}^2 = \frac{1}{n}\sum_{i=1}^n \|f(X_i)\|^2$ where $X_1, \ldots, X_n$ are the empirical samples. We denote $\mathcal{N}(r, \mathcal{F}, L_\infty(P_n))$ as the covering number of $\mathcal{F}$ w.r.t. $L_\infty(P_n)$, which is defined as $\|f\|_{L_\infty(P_n)} = \max_{1 \le i \le n} \|f(X_i)\|$. It is easy to check that

$$\mathcal{N}(r, \mathcal{F}, L_2(P_n)) \le \mathcal{N}(r, \mathcal{F}, L_\infty(P_n)) \le \mathcal{N}(r, \mathcal{F}, \|\cdot\|_\infty).$$

---

[*]Corresponding authors

35th Conference on Neural Information Processing Systems (NeurIPS 2021).

# B Restriction on the domain of uniformly bounded Lipschitz function class $\mathcal{F}^1$

So far, most of the related works assume that the target distribution $\mu$ is supported on a compact set, for example Chen et al. (2020) and Liang (2020). To remove the compact support assumption, we need to assume Assumption 1, i.e., the tails of the target $\mu$ and the reference $\nu$ are subexponential. Define $\mathcal{F}_n^1 := \{f|_{B_2^{d+1}(\sqrt{2}\log n)} : f \in \mathcal{F}^1\}$. In this section, we show that proving Theorem 3.2 is equivalent to establishing the same convergence rate but with the domain restricted function class $\mathcal{F}_n^1$ as the evaluation class.

Under Assumption 1 and by the Markov inequality, we have

$$P_\nu(\|z\| > \log n) \leq \frac{\mathbb{E}_\nu \|z\| \mathbb{1}_{\{\|z\| > \log n\}}}{\log n} = O(n^{-\frac{(\log n)^\delta}{d}} / \log n) \tag{B.1}$$

The Dudley distance between latent joint distribution $\hat{\nu}$ and data joint distribution $\hat{\mu}$ is defined as

$$d_{\mathcal{F}^1}(\hat{\nu}, \hat{\mu}) = \sup_{f \in \mathcal{F}^1} \mathbb{E}f(\hat{g}(z), z) - \mathbb{E}f(x, \hat{e}(x)) \tag{B.2}$$

The first term above can be decomposed as

$$\mathbb{E}f(\hat{g}(z), z) = \mathbb{E}f(\hat{g}(z), z)\mathbb{1}_{\|z\| \leq \log n} + \mathbb{E}f(\hat{g}(z), z)\mathbb{1}_{\|z\| > \log n} \tag{B.3}$$

For any $f \in \mathcal{F}^1$ and fixed point $z_0$ such that $\|z_0\| \leq \log n$, due to the Lipschitzness of $f$, the second term above satisfies

$$\begin{aligned}
|\mathbb{E}f(\hat{g}(z), z)\mathbb{1}_{\|z\| > \log n}| &\leq |\mathbb{E}f(\hat{g}(z), z)\mathbb{1}_{\|z\| > \log n} - \mathbb{E}f(\hat{g}(z_0), z_0)\mathbb{1}_{\|z\| > \log n}| \\
&\quad + |\mathbb{E}f(\hat{g}(z_0), z_0)\mathbb{1}_{\|z\| > \log n}| \\
&\leq \mathbb{E}\|(\hat{g}(z) - \hat{g}(z_0), z - z_0)\|\mathbb{1}_{\|z\| > \log n} + BP_\nu(\|z\| > \log n) \\
&\leq \mathbb{E}(\|(\hat{g}(z) - \hat{g}(z_0)\| + \|z - z_0\|)\mathbb{1}_{\|z\| > \log n} + BP_\nu(\|z\| > \log n) \\
&\leq 2(\log n)P_\nu(\|z\| > \log n) + \mathbb{E}\|z - z_0\|\mathbb{1}_{\|z\| > \log n} + BP_\nu(\|z\| > \log n) \\
&= O(n^{-\frac{(\log n)^\delta}{d}})
\end{aligned}$$

where the second inequality is due to lipschitzness and boundedness of $f$, and the last inequality is due to Assumption 1, (B.1), and the boundedness condition of $\hat{g}$. In the first term in (B.3), $f$ only acts on the increasing $L_2$ ball $B_2^d(\sqrt{2}\log n)$ because of Condition 1 and the indicator function $\mathbb{1}_{\{\|z\| \leq \log n\}}$. Similarly, we can apply the same procedure to the second term in (B.2). Therefore, it is still an equivalent problem if we restrict the domain of $\mathcal{F}^1$ on $B_2^d(\sqrt{2}\log n)$. Hence, in order to prove the estimation error rate in Theorem 3.2, we only need to show that for the restricted evaluation function class $\mathcal{F}_n^1$, we have

$$\mathbb{E}d_{\mathcal{F}_n^1}(\hat{\nu}, \hat{\mu}) \leq C_0 \sqrt{d} n^{-\frac{1}{d+1}}(\log n)^{1+\frac{1}{d+1}} \wedge C_d n^{-\frac{1}{d+1}} \log n$$

Due to this fact, to keep notation simple, we are going to denote $\mathcal{F}_n^1$ as $\mathcal{F}^1$ in the following sections.

**Remark 1.** *The restriction on $\mathcal{F}^1$ is technically necessary for calculating the covering number of $\mathcal{F}^1$ later we will see the use of it when bounding the stochastic error $\mathcal{E}_3$ and $\mathcal{E}_4$ below.*

# C Stochastic errors

## C.1 Bounding $\mathcal{E}_3$ and $\mathcal{E}_4$

The stochastic errors $\mathcal{E}_3$ and $\mathcal{E}_4$ quantify how close the empirical distributions and the true latent joint distribution (data joint distribution) are with the Lipschitz class $\mathcal{F}^1$ as the evaluation class under IPM. We apply the results in Lemma C.1 to bound $\mathcal{E}_3$ and $\mathcal{E}_4$. We introduce two methods to bound $\max\{\mathcal{E}_3, \mathcal{E}_4\}$, which gives two different upper bounds for $\max\{\mathcal{E}_3, \mathcal{E}_4\}$. They both utilize the following lemma, which we shall prove later. More detailed description about the refined Dudley inequality can be found in Srebro and Sridharan (2010) and Schreuder (2020).

**Lemma C.1** (Refined Dudley Inequality). *For a symmetric function class $\mathcal{F}$ with $\sup_{f \in \mathcal{F}} \|f\|_\infty \leq M$, we have*

$$\mathbb{E}[d_{\mathcal{F}}(\hat{\mu}_n, \mu)] \leq \inf_{0 < \delta < M} \left( 4\delta + \frac{12}{\sqrt{n}} \int_\delta^M \sqrt{\log \mathcal{N}(\epsilon, \mathcal{F}, \|\cdot\|_\infty)} \, d\epsilon \right).$$

**Remark 2.** *The original Dudley inequality (Dudley, 1967; Van der Vaart and Wellner, 1996) suffers from the problem that if the covering number $\mathcal{N}(\epsilon, \mathcal{F}, \|\cdot\|_\infty)$ increases too fast as $\epsilon$ goes to $0$, then the upper bound can be infinite. The improved Dudley inequality circumvents this problem by only allowing $\epsilon$ to integrate from $\delta > 0$, which also indicates that $\mathbb{E}\mathcal{E}_3$ scales with the covering number $\mathcal{N}(\epsilon, \mathcal{F}^1, \|\cdot\|_\infty)$.*

### C.1.1 The first method (explicit constant)

The first method provides an explicit constant depending on $d$ at the expense of the higher order of $\log n$ in the upper bounds. It utilizes the next lemma (Gottlieb et al., 2013, Lemma 6), which turns the problem of bounding the covering number of a Lipschitz function class into the one bounding the covering number of the domain defined for the function class.

**Lemma C.2** (Gottlieb et al. (2013)). *Let $\mathcal{F}^L$ be the collection of $L-$Lipschitz functions mapping the metric space $(\mathcal{X}, \rho)$ to $[0, 1]$. Then the covering number of $\mathcal{F}^L$ can be estimated in terms of the covering number of $\mathcal{X}$ with respect to $\rho$ as follows.*

$$\mathcal{N}(\epsilon, \mathcal{F}^L, \|\cdot\|_\infty) \leq \left(\frac{8}{\epsilon}\right)^{\mathcal{N}(\epsilon/8L, \mathcal{X}, \rho)}.$$

Now we apply Lemma C.2 to bound the covering number for the 1-Lipschitz class $\mathcal{N}(\epsilon, \mathcal{F}^1, \|\cdot\|_\infty)$ by bounding the covering number for its domain $\mathcal{N}(\epsilon, B_2^{d+1}(\sqrt{2}\log n), \|\cdot\|_2)$. Define a new function class $\mathcal{F}^{\frac{1}{2B}}$ as

$$\mathcal{F}^{\frac{1}{2B}} := \{\frac{f + B}{2B} : f \in \mathcal{F}^1\}.$$

Recall that $\mathcal{F}^1$ is restricted on $B_2^{d+1}(\sqrt{2}\log n)$. Obviously, $\mathcal{F}^{\frac{1}{2B}}$ is a $\frac{1}{2B}-$Lipschitz function class : $B_2^{d+1}(\sqrt{2}\log n) \mapsto [0, 1]$. A direct application of Lemma C.2 shows that

$$\mathcal{N}(\epsilon, \mathcal{F}^{\frac{1}{2B}}, \|\cdot\|_\infty) \leq \left(\frac{8}{\epsilon}\right)^{\mathcal{N}(\epsilon B/4, B_2^{d+1}(\sqrt{2}\log n), \|\cdot\|_2)}. \tag{C.1}$$

By the definition of $\mathcal{F}^{\frac{1}{2B}}$, the covering numbers satisfy

$$\mathcal{N}(2B\epsilon, \mathcal{F}^1, \|\cdot\|_\infty) = \mathcal{N}(\epsilon, \mathcal{F}^{\frac{1}{2B}}, \|\cdot\|_\infty). \tag{C.2}$$

Note that $B_2^{d+1}(\sqrt{2}\log n)$ is a subset of $[-\sqrt{2}\log n, \sqrt{2}\log n]^d$, and $[-\sqrt{2}\log n, \sqrt{2}\log n]^d$ can be covered with finite $\epsilon$-balls in $\mathbb{R}^d$ that cover the small hypercube with side length $2\epsilon/\sqrt{d}$. It follows that

$$\mathcal{N}(\epsilon, B_2^{d+1}(\sqrt{2}\log n), \|\cdot\|_2) \leq \left(\frac{\sqrt{2(d+1)}\log n}{\epsilon}\right)^{d+1}. \tag{C.3}$$

Combining (C.1), (C.2) and (C.3), we obtain an upper bound for the covering number of the 1-Lipschitz class $\mathcal{F}^1$

$$\log \mathcal{N}(\epsilon, \mathcal{F}^1, \|\cdot\|_\infty) \leq \left(\frac{8\sqrt{2(d+1)}\log n}{\epsilon}\right)^{d+1} \log \frac{16B}{\epsilon}. \tag{C.4}$$

With the upper bound for the covering entropy in (C.4), a direct application of Lemma C.1 (see Section E for details) by taking $\delta = 8\sqrt{2(d+1)}n^{-\frac{1}{d+1}}(\log n)^{1+\frac{1}{d+1}}$ leads to

$$\max\{\mathbb{E}\mathcal{E}_3, \mathbb{E}\mathcal{E}_4\} = O\left(\sqrt{d}n^{-\frac{1}{d+1}}(\log n)^{1+\frac{1}{d+1}} + n^{-\frac{1}{d+1}}(\log n)^{1+\frac{1}{d+1}}\right) \tag{C.5}$$

$$= O\left(\sqrt{d}n^{-\frac{1}{d+1}}(\log n)^{1+\frac{1}{d+1}}\right). \tag{C.6}$$

### C.1.2  The second method (better order of $\log n$)

We now consider the second method that leads to a better order for the $\log n$ term in the upper bound at the expense of explicitness of the constant related to $d$. The next lemma directly provides an upper bound for the covering number of Lipschitz class but with an implicit constant related to $d$. It is a straightforward corollary of Van der Vaart and Wellner (1996, Theorem 2.7.1).

**Lemma C.3.** *Let $\mathcal{X}$ be a bounded, convex subset of $\mathbb{R}^d$ with nonempty interior. There exists a constant $c_d$ depending only on $d$ such that*

$$\log \mathcal{N}(\epsilon, \mathcal{F}^1(\mathcal{X}), \|\cdot\|_\infty) \leq c_d \lambda(\mathcal{X}^1) \left(\frac{1}{\epsilon}\right)^d$$

*for every $\epsilon > 0$, where $\mathcal{F}^1(\mathcal{X})$ is the 1-Lipschitz function class defined on $\mathcal{X}$, and $\lambda(\mathcal{X}^1)$ is the Lebesgue measure of the set $\{x : \|x - \mathcal{X}\| < 1\}$.*

Applying Lemmas C.1 and C.3 (see Section E for details) by taking $\delta = n^{-\frac{1}{d+1}} \log n$ yields

$$\max\{\mathbb{E}\mathcal{E}_3, \mathbb{E}\mathcal{E}_4\} = O\left(C_d n^{-\frac{1}{d+1}} \log n\right), \tag{C.7}$$

where $C_d$ is some constant depending on $d$. Combining (C.6) and (C.7), we get

$$\max\{\mathbb{E}\mathcal{E}_3, \mathbb{E}\mathcal{E}_4\} = O\left(C_d n^{-\frac{1}{d+1}} \log n \wedge \sqrt{d} n^{-\frac{1}{d+1}} (\log n)^{1+\frac{1}{d+1}}\right). \tag{C.8}$$

**Remark 3.** *Here, we have a tradeoff between the logarithmic factor $\log n$ and the explicitness of the constant depending on $d$. If we want an explicit constant depending on $d$, then we have the factor $(\log n)^{1+\frac{1}{d+1}}$ in the upper bound. Later we will see that $\mathbb{E}\mathcal{E}_3$ and $\mathbb{E}\mathcal{E}_4$ are the dominating terms in the four error terms, hence the explicitness of the corresponding constant becomes important. Therefore, we list two different methods here to bound $\mathbb{E}\mathcal{E}_3$ and $\mathbb{E}\mathcal{E}_4$.*

### C.2  Combination of the four error terms

With all the upper bounds for the four different error terms obtained above, next we consider $\mathcal{E}_1$-$\mathcal{E}_4$ simultaneously to obtain an overall convergence rate. First, recall how we bound $\mathcal{E}_1$ and $\mathcal{E}_4$. With Lemma 4.2, we have

$$\mathcal{E}_1 = O\left(\sqrt{d}(W_1 L_1)^{-\frac{2}{d+1}} \log n\right). \tag{C.9}$$

To control $\mathcal{E}_1$ while keeping the architecture of discriminator class $\mathcal{F}_{NN}$ as small as possible, we let $W_1 L_1 = \lceil \sqrt{n} \rceil$, so that $\mathcal{E}_1 = O\left(\sqrt{d} n^{-\frac{1}{d+1}} \log n\right)$ dominated by $\mathcal{E}_3$ and $\mathcal{E}_4$.

By Theorem 4.3, we can choose the architectures of generator and encoder classes accordingly to perfectly control $\mathcal{E}_2$, i.e. $\mathcal{E}_2 = 0$.

We note that because we imposed Condition 1 on both generator and encoder classes, Theorem 4.3 can not be applied if we have some $\|x_i\|$ or $\|z_i\|$ greater than $\log n$, in which case $\mathcal{E}_2$ can not be perfectly controlled. But we can still handle this case by considering the probability of the bad set.

Under Condition 1, on the nice set $A := \{\max_{1 \leq i \leq n} \|x_i\| \leq \log n\} \cap \{\max_{1 \leq i \leq n} \|z_i\| \leq \log n\}$, we have $\mathcal{E}_2 = 0$. Probability of the nice set $A$ has the following lower bound.

$$P(A) = P_\mu(\|x_i\| \leq \log n)^n \cdot P_\nu(\|z_i\| \leq \log n)^n$$
$$\geq \left(1 - C n^{-\frac{(\log n)^\delta}{d}}\right)^{2n}, \text{ for some constant } C > 0 \text{ by Assumption 1}$$
$$\geq 1 - C n^{-\frac{(\log n)^\delta}{d}} \cdot (2n), \text{ for large } n.$$

The bad set $A^c$ is where $\mathcal{E}_2 > 0$, which has the probability upper bound as follows.

$$P(A^c) \leq C n^{-\frac{(\log n)^\delta}{d}} \cdot (2n)$$
$$= O\left(n^{-\frac{(\log n)^{\delta'}}{d}}\right), \text{ for any } \delta' < \delta.$$

In Assumption 1, the $(\log n)^\delta$ factor was to make the tail of the target $\mu$ strictly subexponential, which leads to $P(A^c) \to 0$, while the exponential tail or heavier will cause the undesired result $P(A^c) \to 1$.

Now we are ready to obtain the desired result in Theorem 3.2. The nice set $A = \{\max_{1 \le i \le n} \|x_i\| \le \log n\} \cap \{\max_{1 \le i \le n} \|z_i\| \le \log n\}$ is where $\mathcal{E}_2 = 0$. By combining the results discussed above, we have

$$
\begin{aligned}
\mathbb{E} d_{\mathcal{F}^1}(\hat{\nu}, \hat{\mu}) &= 2\mathcal{E}_1 + \mathcal{E}_2 \mathbb{1}_A + \mathcal{E}_2 \mathbb{1}_{A^c} + \mathbb{E}\mathcal{E}_3 + \mathbb{E}\mathcal{E}_4 \\
&\le O\left( \sqrt{d} n^{-\frac{1}{d+1}} \log n + 0 + 2B P_\mu(A^c) + \sqrt{d} n^{-\frac{1}{d+1}} (\log n)^{1+\frac{1}{d+1}} \wedge C_d n^{-\frac{1}{d+1}} \log n \right) \\
&= O\left( \sqrt{d} n^{-\frac{1}{d+1}} (\log n)^{1+\frac{1}{d+1}} \wedge C_d n^{-\frac{1}{d+1}} \log n + n^{-\frac{(\log n)^{\delta'}}{d}} \right) \\
&= O\left( \sqrt{d} n^{-\frac{1}{d+1}} (\log n)^{1+\frac{1}{d+1}} \wedge C_d n^{-\frac{1}{d+1}} \log n \right),
\end{aligned}
$$

which completes the proof of Theorem 3.2.

# D  Proof of Inequality (4.2)

For ease of reference, we restate inequality (4.2) as the following lemma.

**Lemma 4.2.** *For any symmetric function classes $\mathcal{F}$ and $\mathcal{H}$, denote the approximation error $\mathcal{E}(\mathcal{H}, \mathcal{F})$ as*

$$
\mathcal{E}(\mathcal{H}, \mathcal{F}) := \sup_{h \in \mathcal{H}} \inf_{f \in \mathcal{F}} \|h - f\|_\infty,
$$

*then for any probability distributions $\mu$ and $\nu$,*

$$
d_{\mathcal{H}}(\mu, \nu) - d_{\mathcal{F}}(\mu, \nu) \le 2\mathcal{E}(\mathcal{H}, \mathcal{F}).
$$

*Proof of Lemma 4.2.* By the definition of supremum, for any $\epsilon > 0$, there exists $h_\epsilon \in \mathcal{H}$ such that

$$
\begin{aligned}
d_{\mathcal{H}}(\mu, \nu) :&= \sup_{h \in \mathcal{H}} [\mathbb{E}_\mu h - \mathbb{E}_\nu h] \\
&\le \mathbb{E}_\mu h_\epsilon - \mathbb{E}_\nu h_\epsilon + \epsilon \\
&= \inf_{f \in \mathcal{F}} [\mathbb{E}_\mu(h_\epsilon - f) - \mathbb{E}_\nu(h_\epsilon - f) + \mathbb{E}_\mu(f) - \mathbb{E}_\nu(f)] + \epsilon \\
&\le 2 \inf_{f \in \mathcal{F}} \|h_\epsilon - f\|_\infty + d_{\mathcal{F}}(\mu, \nu) + \epsilon \\
&\le 2\mathcal{E}(\mathcal{H}, \mathcal{F}) + d_{\mathcal{F}}(\mu, \nu) + \epsilon,
\end{aligned}
$$

where the last line is due to the definition of $\mathcal{E}(\mathcal{H}, \mathcal{F})$. $\qquad\square$

It is easy to check that if we replace $d_{\mathcal{H}}(\mu, \nu)$ by $\hat{d}_{\mathcal{H}}(\mu, \nu) := \sup_{h \in \mathcal{H}} [\hat{\mathbb{E}}_\mu h - \hat{\mathbb{E}}_\nu h]$, Lemma 4.2 still holds.

# E  Bounding $\mathbb{E}\mathcal{E}_3$ and $\mathbb{E}\mathcal{E}_4$

## E.1  Method One

With the upper bound for the covering entropy (C.4), i.e.

$$
\log \mathcal{N}(\epsilon, \mathcal{F}^1, \|\cdot\|_\infty) \le \left( \frac{8\sqrt{2(d+1)} \log n}{\epsilon} \right)^{d+1} \log \frac{16B}{\epsilon}
$$

and $\delta = 8\sqrt{2(d+1)}n^{-\frac{1}{d+1}}(\log n)^{1+\frac{1}{d+1}}$, applying Lemma C.1 we have

$$
\mathbb{E}\mathcal{E}_3 = O\left(\delta + n^{-\frac{1}{2}}\int_\delta^B \left(\frac{8\sqrt{2(d+1)}\log n}{\epsilon}\right)^{\frac{d+1}{2}}\left(\log\frac{16B}{\epsilon}\right)^{\frac{1}{2}}d\epsilon\right)
$$

$$
= O\left(\delta + n^{-\frac{1}{2}}(8\sqrt{2(d+1)}\log n)^{\frac{d+1}{2}}(\frac{\log n}{d+1})^{\frac{1}{2}}\delta^{1-\frac{d+1}{2}}\right)
$$

$$
= O\left(\sqrt{d}n^{-\frac{1}{d+1}}(\log n)^{1+\frac{1}{d+1}} + n^{-\frac{1}{d+1}}(\log n)^{1+\frac{1}{d+1}}\right)
$$

$$
= O\left(\sqrt{d}n^{-\frac{1}{d+1}}(\log n)^{1+\frac{1}{d+1}}\right),
$$

where the second equality is due to

$$
\log\frac{16B}{\epsilon} = O\left(\log\frac{1}{\epsilon}\right) = O\left(\log\left(\frac{n^{\frac{1}{d+1}}}{8\sqrt{2(d+1)}(\log n)^{1+\frac{1}{d+1}}}\right)\right) = O\left(\log n^{\frac{1}{d+1}}\right),
$$

and the third equality follows from simple algebra.

### E.2  Method Two

By Lemma C.3, we have

$$
\log\mathcal{N}(\epsilon,\mathcal{F}^1,\|\cdot\|_\infty) \le c_d\left(\frac{\log n}{\epsilon}\right)^{d+1}.
$$

Taking $\delta = n^{-\frac{1}{d+1}}\log n$ and applying Lemma C.1, we obtain

$$
\mathbb{E}\mathcal{E}_3 = O\left(\delta + (\frac{c_d}{n})^{\frac{1}{2}}(\log n)^{\frac{d+1}{2}}\int_\delta^M(\frac{1}{\epsilon})^{\frac{d+1}{2}}d\epsilon\right)
$$

$$
= \tilde{O}\left(\delta + n^{-\frac{1}{2}}(\log n)^{\frac{d+1}{2}}\delta^{1-\frac{d+1}{2}}\right)
$$

$$
= \tilde{O}\left(n^{-\frac{1}{d+1}}\log n\right),
$$

where $\tilde{O}(\cdot)$ omitted the constant related to $d$.

## F  Proof of Lemma C.1

For completeness we provide a proof of the refined Dudley's inequality in Lemma C.1. We apply the standard symmetrization and chaining technics in the proof, see, for example, Van der Vaart and Wellner (1996).

*Proof.* Let $Y_1,\ldots,Y_n$ be random samples from $\mu$ which are independent of $X_i's$. Then we have

$$
\mathbb{E}d_\mathcal{F}(\hat{\mu}_n,\mu) = \mathbb{E}\sup_{f\in\mathcal{F}}[\frac{1}{n}\sum_{i=1}^n f(X_i) - \mathbb{E}f(X_i)]
$$

$$
= \mathbb{E}\sup_{f\in\mathcal{F}}[\frac{1}{n}\sum_{i=1}^n f(X_i) - \mathbb{E}\frac{1}{n}\sum_{i=1}^n f(Y_i)]
$$

$$
\le \mathbb{E}_{X,Y}\sup_{f\in\mathcal{F}}[\frac{1}{n}\sum_{i=1}^n f(X_i) - \frac{1}{n}\sum_{i=1}^n f(Y_i)]
$$

$$
= \mathbb{E}_{X,Y}\sup_{f\in\mathcal{F}}[\frac{1}{n}\sum_{i=1}^n \epsilon_i(f(X_i) - f(Y_i))]
$$

$$
\le 2\mathbb{E}\hat{\mathcal{R}}_n(\mathcal{F})
$$

where the first inequality is due to Jensen inequality, and the third equality is because that $(f(X_i) - f(Y_i))$ has symmetric distribution.

Let $\alpha_0 = M$ and for any $j \in \mathbb{N}_+$ let $\alpha_j = 2^{-j}M$. For each $j$, let $T_i$ be a $\alpha_i$-cover of $\mathcal{F}$ w.r.t. $L_2(P_n)$ such that $|T_i| = \mathcal{N}(\alpha_i, \mathcal{F}, L_2(P_n))$. For each $f \in \mathcal{F}$ and $j$, pick a function $\hat{f}_i \in T_i$ such that $\|\hat{f}_i - f\|_{L_2(P_n)} < \alpha_i$. Let $\hat{f}_0 = 0$ and for any $N$, we can express $f$ by chaining as

$$f = f - \hat{f}_N + \sum_{i=1}^{N}(\hat{f}_i - \hat{f}_{i-1}).$$

Hence for any $N$, we can express the empirical Rademacher complexity as

$$\hat{\mathcal{R}}_n(\mathcal{F}) = \frac{1}{n}\mathbb{E}_\epsilon \sup_{f \in \mathcal{F}} \sum_{i=1}^{n} \epsilon_i \left( f(X_i) - \hat{f}_N(X_i) + \sum_{j=1}^{N}(\hat{f}_j(X_i) - \hat{f}_{j-1}(X_i)) \right)$$

$$\leq \frac{1}{n}\mathbb{E}_\epsilon \sup_{f \in \mathcal{F}} \sum_{i=1}^{n} \epsilon_i \left( f(X_i) - \hat{f}_N(X_i) \right) + \sum_{i=1}^{n} \frac{1}{n}\mathbb{E}_\epsilon \sup_{f \in \mathcal{F}} \sum_{j=1}^{N} \epsilon_i \left( \hat{f}_j(X_i) - \hat{f}_{j-1}(X_i) \right)$$

$$\leq \|\epsilon\|_{L_2(P_n)} \sup_{f \in \mathcal{F}} \|f - \hat{f}_N\|_{L_2(P_n)} + \sum_{i=1}^{n} \frac{1}{n}\mathbb{E}_\epsilon \sup_{f \in \mathcal{F}} \sum_{j=1}^{N} \epsilon_i \left( \hat{f}_j(X_i) - \hat{f}_{j-1}(X_i) \right)$$

$$\leq \alpha_N + \sum_{i=1}^{n} \frac{1}{n}\mathbb{E}_\epsilon \sup_{f \in \mathcal{F}} \sum_{j=1}^{N} \epsilon_i \left( \hat{f}_j(X_i) - \hat{f}_{j-1}(X_i) \right),$$

where $\epsilon = (\epsilon_1, \ldots, \epsilon_n)$ and the second-to-last inequality is due to Cauchy–Schwarz. Now the second term is the summation of empirical Rademacher complexity w.r.t. the function classes $\{f' - f'' : f' \in T_j, f'' \in T_{j-1}\}, j = 1, \ldots, N$. Note that

$$\|\hat{f}_j - \hat{f}_{j-1}\|_{L_2(P_n)}^2 \leq \left( \|\hat{f}_j - f\|_{L_2(P_n)} + \|f - \hat{f}_{j-1}\|_{L_2(P_n)} \right)^2$$

$$\leq (\alpha_j + \alpha_{j-1})^2$$

$$= 3\alpha_j^2.$$

Massart's lemma (Mohri et al., 2018, Theorem 3.7) states that if for any finite function class $\mathcal{F}$, $\sup_{f \in \mathcal{F}} \|f\|_{L_2(P_n)} \leq M$, then we have

$$\hat{\mathcal{R}}_n(\mathcal{F}) \leq \sqrt{\frac{2M^2 \log(|\mathcal{F}|)}{n}}.$$

Applying Massart's lemma to the function classes $\{f' - f'' : f' \in T_j, f'' \in T_{j-1}\}, j = 1, \ldots, N$, we get that for any $N$,

$$\hat{\mathcal{R}}_n(\mathcal{F}) \leq \alpha_N + \sum_{j=1}^{N} 3\alpha_j \sqrt{\frac{2 \log(|T_j| \cdot |T_{j-1}|)}{n}}$$

$$\leq \alpha_N + 6 \sum_{j=1}^{N} \alpha_j \sqrt{\frac{\log(|T_j|)}{n}}$$

$$\leq \alpha_N + 12 \sum_{j=1}^{N} (\alpha_j - \alpha_{j+1}) \sqrt{\frac{\log \mathcal{N}(\alpha_j, \mathcal{F}, L_2(P_n))}{n}}$$

$$\leq \alpha_N + 12 \int_{\alpha_{N+1}}^{\alpha_0} \sqrt{\frac{\log \mathcal{N}(r, \mathcal{F}, L_2(P_n))}{n}} dr,$$

where the third inequality is due to $2(\alpha_j - \alpha_{j+1}) = \alpha_j$. Now for any small $\delta > 0$ we can choose $N$ such that $\alpha_{N+1} \leq \delta < \alpha_N$. Hence,

$$\hat{\mathcal{R}}_n(\mathcal{F}) \leq 2\delta + 12 \int_{\delta/2}^{M} \sqrt{\frac{\log \mathcal{N}(r, \mathcal{F}, L_2(P_n))}{n}} dr.$$

Since $\delta > 0$ is arbitrary, we can take $\inf$ w.r.t. $\delta$ to get

$$\hat{\mathcal{R}}_n(\mathcal{F}) \leq \inf_{0 < \delta < M} \left( 4\delta + 12 \int_\delta^M \sqrt{\frac{\log \mathcal{N}(r, \mathcal{F}, L_2(P_n))}{n}} \, dr \right).$$

The result follows due to the fact that

$$\mathcal{N}(r, \mathcal{F}, L_2(P_n)) \leq \mathcal{N}(\epsilon, \mathcal{F}, L_\infty(P_n)) \leq \mathcal{N}(\epsilon, \mathcal{F}, \|\cdot\|_\infty).$$

$\square$

# G    Proof of Theorem 3.1

*Proof.* Taking $W_1 L_1 = \lceil \sqrt{n} \rceil$, Shen et al. (2019, Theorem 4.3) gives rise to $\mathcal{E}_1 = O(\sqrt{d} n^{-\frac{1}{d+1}})$. The range of $g$ and $e$ covers the supports of $\mu$ and $\nu$, respectively, hence Theorem 4.3 leads to $\mathcal{E}_2 = 0$. By Lemma C.2, we have

$$\log \mathcal{N}(\epsilon, \mathcal{F}^1, \|\cdot\|_\infty) \leq \left( \frac{8\sqrt{2(d+1)}M}{\epsilon} \right)^{d+1} \log \frac{16B}{\epsilon}.$$

Now following the same procedure as in Section E by taking $\delta = 8\sqrt{2(d+1)} n^{-\frac{1}{d+1}} (\log n)^{\frac{1}{d+1}}$, we have

$$\max\{\mathbb{E}\mathcal{E}_3, \mathbb{E}\mathcal{E}_4\} = O\left( \sqrt{d} n^{-\frac{1}{d+1}} (\log n)^{\frac{1}{d+1}} \right).$$

At last, we consider all four error terms simultaneously.

$$\begin{aligned}
\mathbb{E} d_{\mathcal{F}^1}(\hat{\nu}, \hat{\mu}) &\leq \mathcal{E}_1 + \mathcal{E}_2 + \mathbb{E}\mathcal{E}_4 + \mathbb{E}\mathcal{E}_3 \\
&= O(\sqrt{d} n^{-\frac{1}{d+1}} + 0 + \sqrt{d} n^{-\frac{1}{d+1}} (\log n)^{\frac{1}{d+1}}) \\
&= O(\sqrt{d} n^{-\frac{1}{d+1}} (\log n)^{\frac{1}{d+1}}).
\end{aligned}$$

$\square$

# H    Proof of Theorem 3.3

Following the same proof as Theorem 4.3, we have the following theorem.

**Theorem H.1.** *Suppose $\nu$ supported on $\mathbb{R}^k$ and $\mu$ supported on $\mathbb{R}^d$ are both absolutely continuous w.r.t. Lebesgue measure, and $z_i's$ and $x_i's$ are i.i.d. samples from $\nu$ and $\mu$, respectively for $1 \leq i \leq n$. Then there exist generator and encoder neural network functions $g : \mathbb{R}^k \mapsto \mathbb{R}^d$ and $e : \mathbb{R}^d \mapsto \mathbb{R}^k$ such that $g$ and $e$ are inverse bijections of each other between $\{z_i : 1 \leq i \leq n\}$ and $\{x_i : 1 \leq i \leq n\}$. Moreover, such neural network functions $g$ and $e$ can be obtained by properly specifying $W_2^2 L_2 = c_2 dn$ and $W_3^2 L_3 = c_3 kn$ for some constant $12 \leq c_2, c_3 \leq 384$.*

Since $\mu$ and $\nu$ are absolutely continuous by assumption, they are also absolutely continuous in any one dimension. Hence the proof reduces to the one-dimensional case.

# I    Additional Lemma

Denote $\mathcal{S}^d(z_0, \ldots, z_{N+1})$ as the set of all continuous piecewise linear functions $f : \mathbb{R} \mapsto \mathbb{R}^d$ which have breakpoints only at $z_0 < z_1 < \cdots < z_N < z_{N+1}$ and are constant on $(-\infty, z_0)$ and $(z_{N+1}, \infty)$. The following lemma is a result in Yang et al. (2021).

**Lemma I.1.** *Suppose that $W \geq 7d + 1$, $L \geq 2$ and $N \leq (W - d - 1) \lfloor \frac{W-d-1}{6d} \rfloor \lfloor \frac{L}{2} \rfloor$. Then for any $z_0 < z_1 < \cdots < z_N < z_{N+1}$, $\mathcal{S}^d(z_0, \ldots, z_{N+1})$ can be represented by a ReLU FNNs with width and depth no larger than $W$ and $L$, respectively.*

This result indicates that the expressive capacity of ReLU FNNs for piecewise linear functions. If we choose $N = (W - d - 1) \lfloor \frac{W-d-1}{6d} \rfloor \lfloor \frac{L}{2} \rfloor$, a simple calculation shows $cW^2 L/d \leq N \leq CW^2 L/d$ with $c = 1/384$ and $C = 1/12$. This means when the number of breakpoints are moderate compared with the network structure, such piecewise linear functions are expressible by feedforward ReLU networks.