# OpenReview forum: "Non-asymptotic Error Bounds for Bidirectional GANs"
_NeurIPS.cc/2021/Conference — NeurIPS 2021 Poster_

### Official Review · Reviewer_jF5w · 2021-07-16

**Rating:** 6
**Confidence:** 3

**Summary:**

This paper presents non-asymptotic error bounds for bidirectional GAN estimators. Their primary contribution is a dependence on the square root of dimension under certain assumptions on the data distribution and the latent distribution.

**Limitations And Societal Impact:**

The authors discuss some societal impact, but there are no societal issues due to the theoretical nature of this work.

**Main Review:**

1. Originality

The task is not new, but the results are new. One minor comment for highlighting differences with previous contributions is to write out previously known results in terms of equations, rather than only in plain words as is the case of the current version. Related work is adequately cited.

2. Quality

The submission seems technically sound. It might be a good idea to add a bit more discussion into the dependence on n and C_d as well---the current paper focuses on the square root dependence on d, but the main theorems (3.1., 3.2., 3.3.) all depend on specifying generator and discriminator class structure in terms of n, d., and C_d "depends on d". I think it would be helpful to have some more discussion in terms of these dependencies. Following the above point, it would be nice to address whether general results would also hold without a specification of the class structures; otherwise, the claims seem to be fairly specific and only holds for these particular specifications.

3. Clarity

The paper is fairly well-written and organized. Some comments on typos are addressed in Section 5. of the main review. Some comments regarding clarity:

a. The notation $\hat{\mathbb{E}}$ is not clearly defined in Line 188-189.

b. There is no notion of specifying network structures in the abstract, and it leaves an impression that the results are general and hold true for a wide range of bidrectional GANs. This may need to be addressed in further revisions of this paper.

c. Following the above point, it leads to the question of whether the assumptions and conditions are suitable for this particular specification, and what would happen under looser assumptions: even in Section 2.2 the authors pointed out that "Assumptions 1 and 2 can be easily satisfied by specifying $\nu$ as some common distribution with easy-to-sample density such as Gaussian or uniform...". It would be interesting if the necessity of these assumptions are discussed in a clearer sense.

d. The use of the $\implies$ signs in Lines 137-139 are slightly ambiguous.

e. Perhaps a minor point, but in the abstract and Section 1 the term "target distribution" is used but is not clearly defined. It may be helpful to clarify in further revisions the difference/similarities of the "target distribution" compared with the "data distribution" and "latent distribution".


4. Significance

The results are interesting and bring insight into understanding bidirectional GANs. From this perspective, the results are important. However, I think the significance would be clearer if some of the above points are addressed.

5. Typos

Some typos:

a. In Theorems 3.2. and 3.3., "Conditions 1" may be "Condition 1".

b. In Line 145: "defined" may be "define".

Update after reviewing comments: Increased score to 6.

**Time Spent Reviewing:**

3 hours

---

> ### Author Response · Authors · 2021-08-10
> **Response to the comments by Reviewer jF5w**
>
> We are very grateful to you for your taking the time to read our paper and for your helpful comments and constructive suggestions.
>
> Main Review:
>
> 1. Originality
>
> The task is not new, but the results are new. One minor comment for highlighting differences with previous contributions is to write out previously known results in terms of equations, rather than only in plain words as is the case of the current version. Related work is adequately cited.
>
> We appreciate your noticing the merits of our paper. We will highlight the differences from the previous contributions more clearly by using equations in the revision.
>
> 2. Quality
>
> The submission seems technically sound. It might be a good idea to add a bit more discussion into the dependence on n and C_d as well---the current paper focuses on the square root dependence on d, but the main theorems (3.1., 3.2., 3.3.) all depend on specifying generator and discriminator class structure in terms of n, d., and C_d "depends on d". I think it would be helpful to have some more discussion in terms of these dependencies. Following the above point, it would be nice to address whether general results would also hold without a specification of the class structures; otherwise, the claims seem to be fairly specific and only holds for these particular specifications.
>
> We will add more detailed discussions about the dependence of the error bound on $n$ and $C_d$. Here we used two methods bounding statistical errors, which led to two different bounds: one with explicit square root d prefactor with the cost that we have an additional   $\log n$ factor;  Another one has an implicit prefactor but without the   $\log n$ factor.  The overall bound is the minimum of the two. Hence, there is a trade-off between the explicitness of prefactor and the order of $\log n$. We will describe this more clearly.
>
> 3. Clarity
>
> The paper is fairly well-written and organized. Some comments on typos are addressed in Section 5. of the main review. Some comments regarding clarity:
>
> Thank you so much for your positive feedback. We will correct the typos and improve the clarity of the paper in the revision.
>
>  a. The notation $\hat{\mathbb{E}}$  is not clearly defined in Line 188-189.
>
> We will clearly define this notation in the revision. It means taking expectation under the empirical distribution.
>
>  b. There is no notion of specifying network structures in the abstract, and it leaves an impression that the results are general and hold true for a wide range of bidirectional GANs. This may need to be addressed in further revisions of this paper.
>
> Thank you very much for pointing this out, we will make the specification of the network structure clear in the revision. Briefly, any ReLU network with the same order depth and width will work. We will also discuss Assumptions 1 and 2 more clearly in the revision. We will make this point clear in the revision.
>
>  c. Following the above point, it leads to the question of whether the assumptions and conditions are suitable for this particular specification, and what would happen under looser assumptions: even in Section 2.2 the authors pointed out that "Assumptions 1 and 2 can be easily satisfied by specifying  as some common distribution with easy-to-sample density such as Gaussian or uniform...". It would be interesting if the necessity of these assumptions are discussed in a clearer sense.
>
> We will have a more detailed discussion about the necessity of the conditions and assumptions. Assumption 1 (tail probability assumption) is necessary to relax the bounded support restriction. Without the tail probability condition and boundedness restriction, the discriminator approximation error would be impossible to bound. Assumption 2 (absolute continuity) is necessary for eliminating the generator and encoder approximation errors and allowing target and source to be of different dimensions. We will add these clarifications in the revision.
>
>  d. The use of the  signs in Lines 137-139 are slightly ambiguous.
>
>  We will change the use of this symbol in Lines 137-139 to remove any ambiguity in the revision.
>
>  e. Perhaps a minor point, but in the abstract and Section 1 the term "target distribution" is used but is not clearly defined. It may be helpful
>  to clarify in further revisions the difference/similarities of the "target distribution" compared with the "data distribution" and "latent
>  distribution".
>
> We will clearly define the terms "target distribution’’, "data distribution’’ and "source distribution’’ in the revision.
>
> 4. Significance
>
> The results are interesting and bring insight into understanding bidirectional GANs. From this perspective, the results are important. However, I think the significance would be clearer if some of the above points are addressed.
>
> Thank you so much for the helpful comments and suggestions. We will carefully revise the paper accordingly.
>
> 5. Typos
>
> Some typos:
>
> a. In Theorems 3.2. and 3.3., "Conditions 1" may be "Condition 1".
>
> We will change "Conditions 1’’ to ``Condition 1’’.
>
> b. In Line 145: "defined" may be "define".
>
> We will change "defined’’ to "define’’.

---

> > ### Comment · Reviewer_jF5w · 2021-08-25
> > **Update after reading comments**
> >
> > Thanks for the detailed replies to the comments. I am increasing my score to 6. As mentioned in the earlier revision, there are some parts that require clarification and it would be beneficial to address them clearly in later versions of the paper.

---

> > > ### Author Response · Authors · 2021-08-25
> > > **Thank you for the update**
> > >
> > > Thank you so much for your updated comments and for your positive feedback.
> > >
> > > We greatly appreciate your work reviewing our paper and your helpful comments and constructive suggestions. We will revise our paper carefully according to your comments and suggestions. We believe your review helped us improve the paper and we are grateful to you for your review.

---

> ### Author Response · Authors · 2021-08-23
> **Any additional comments on our response to your original review?**
>
> Dear reviewer,
>
> Please let us know if you have any additional comments on our response to your original review to help us further improve our paper.
> We really appreciate your work in reviewing our paper!

---

### Official Review · Reviewer_jcaY · 2021-07-18

**Rating:** 6
**Confidence:** 4

**Summary:**

This paper studies the statistical properties of bidirectional GANs. It studies the sample complexity of the bidirectional GAN estimate under the Dudley metric. They relax a couple of assumptions usually made in GAN theory literature; they do not assume that the dimension of the latent distribution and the data distribution are the same or that the true data distribution has bounded support.

**Limitations And Societal Impact:**

The authors have discussed some limitations of this work.

**Main Review:**

This is an interesting paper that provides useful insight for understanding bidirectional GANs. It is clearly written and well organized. This work shows to theoretically handle the bidirectionality in the particular GAN framework and unboundedness of the support of the data. They also show via a decomposition of the generalization error how the different dimensions of the latent and the data distribution affect this error. Moreover, they provide explicit prefactors for the upper bound of the generalization error.

It is important to note that explicit prefactors are only provided in the case of bounded support of the data. In this case the Dudley metric and the Wasserstein distance is the same. Since the paper by Lu and Lu (2020) cited in the related work study the Wasserstein distance, does their proof imply the same dependence of the pre factor on the dimension? Moreover, given that the generalization bound has exponential dependence on the dimension when is a better estimate of the prefactor important?

My other concerns/suggestions with this work are as follows:
- It would be nice to have lower bounds for statistical complexity under the setting considered in this work to better understand the results.
- Often the number of samples from the latent distribution used to train GANs far exceeds that of the true data. It would provide useful perspective to incorporate this into the results.
- Some discussion on the computational complexity of computing the GAN estimator proposed would be very beneficial even if only under the setting where the discriminator is replaced by an oracle that provides the loss.

**Time Spent Reviewing:**

3 hours

---

> ### Author Response · Authors · 2021-08-10
> **Response to the comments by Reviewer jcaY**
>
> We are very grateful to you for your taking the time to read our paper and for your helpful comments and constructive suggestions.
>
>
> Main Review:
> This is an interesting paper that provides useful insight for understanding bidirectional GANs. It is clearly written and well organized. This work shows to theoretically handle the bidirectionality in the particular GAN framework and unboundedness of the support of the data. They also show via a decomposition of the generalization error how the different dimensions of the latent and the data distribution affect this error. Moreover, they provide explicit prefactors for the upper bound of the generalization error.
>
> Thank you very much for noticing the merits of our paper.
>
> It is important to note that explicit prefactors are only provided in the case of bounded support of the data. In this case the Dudley metric and the Wasserstein distance is the same. Since the paper by Lu and Lu (2020) cited in the related work study the Wasserstein distance, does their proof imply the same dependence of the pre factor on the dimension? Moreover, given that the generalization bound has exponential dependence on the dimension when is a better estimate of the prefactor important?
>
> “It is important to note that explicit prefactors are only provided in the case of bounded support of the data”--- We used two methods bounding statistical errors, which leaded to two different bounds: one with explicit square root d prefactor with the cost that we have an additional   $\log n$ factor;  Another one with implicit prefactor but without the   $\log n$ factor.  Hence, it is a tradeoff between the explicitness of prefactor and the order of $\log n$. In the revision, we will clarify this point in the revision.
>
>
> “Since the paper by Lu and Lu (2020) cited in the related work study the Wasserstein distance, does their proof imply the same dependence of the pre factor on the dimension?”--- In Lu and Lu (2020), they only considered the vanilla GAN when target and source have the same dimension by using optimal transport theory.  They mentioned that the square root d factor will appear in sub-gaussian case which is slightly stricter than our assumption 1.  They impose a sub-gaussian moment condition and then apply a triangle inequality given in Section 5.2 in Jing Lei (2020). In our Assumption 1, we only require a sub-exponential which is weaker than the sub-Gaussian condition of Lu and Lu (2020). We will clarify this point in the revision.
>
> “When is a better estimate of the prefactor important” –-- In high dimensional case, where d is large, the prefactor is very important.
>
> My other concerns/suggestions with this work are as follows:
>
> It would be nice to have lower bounds for statistical complexity under the setting considered in this work to better understand the results.
>
> Our bound is tight up to a logarithmic factor, since the minimax rate measured in Wasserstein distance for learning distributions is $\mathcal{O}(n^{-1/d})$, see, for example, Liang (2018). We will comment on this point in the revision.
>
> Often the number of samples from the latent distribution used to train GANs far exceeds that of the true data. It would provide useful perspective to incorporate this into the results.
>
> In GANs, we can make use of more samples to lower both generator and discriminator approximation errors. However, in BiGANs, the bidirectional structure forces us to have equal amount samples otherwise it would be hard to bound both generator and encoder approximation errors. This is an important and interesting question and deserves to be studied carefully in the future.
>
> Some discussion on the computational complexity of computing the GAN estimator proposed would be very beneficial even if only under the setting where the discriminator is replaced by an oracle that provides the loss.
>
> Thank you very much for pointing this out. We agree computational complexity in BiGANs is an important issue.  In this paper our focus is on the error analysis of the BiGAN estimator. We will include some discussions on the computational complexity for BiGANs in the revision.

---

> > ### Comment · Reviewer_jcaY · 2021-08-27
> > **Acknowledge of Authors' Response**
> >
> > Thank you authors for you clear and thorough response. It was sufficient to address the concerns I raised.

---

> > > ### Author Response · Authors · 2021-08-29
> > > **Thank you for the additional feedback**
> > >
> > > Thank you so much for your positive feedback and for your taking the time to review our paper. We are very grateful to you for your review that helps us improve our paper.

---

> ### Author Response · Authors · 2021-08-24
> **Please let us know if you have any additional comments**
>
> Thank so much for your work reviewing our paper and for your helpful comments and constructive suggestions.
> Your review helped us improve the paper and we are very grateful to you for that.
> Please let us know if you have any additional comments after seeing our response so that we can further improve our paper.

---

### Official Review · Reviewer_eow5 · 2021-07-18

**Rating:** 6
**Confidence:** 3

**Summary:**

The paper presents non-asymptotic error analysis for bidirectional GANs. The distinctive feature of the error analysis is relaxing the assumption that the data and latent variable should have the same dimension.  The constants in the error bound is also explicit in dimension.

**Limitations And Societal Impact:**

Yes

**Main Review:**

The paper is well written and discusses the related literature well. I could not go through the details of error analysis, but the statement of the results is rigorous. I have some question/comments about the assumptions and results.


1- What is the motivation to study bi directional GAN as opposed to GAN? Can these results be extended to GAN, or is there a special property of GAN that allows relaxing the assumptions?

2- What are the limitations of Assumption 2? Does it exclude data distributions that lie in low-dimensional manifolds? How necessary is this assumption?

3- For Theorem 3.2., the constant for the unbounded case does not seem to be explicit as the paper suggests. It seems to better distinguish this in the introduction

4- It is surprising that the dimension k does not appear in result in Theorem 3.3. Is there any explanations for this?

5- I did not understand how the results in Theorem 4.3 is extended to multi-dimensional case for $\nu$.

Overall, the paper is well written and comparison is given with the related literature. I only suggest including an outline to navigate the theoretical results

**Time Spent Reviewing:**

3

---

> ### Author Response · Authors · 2021-08-10
> **Response to comments by Reviewer eow5**
>
> We are very grateful to you for your taking the time to read our paper and for your helpful comments and constructive suggestions.
>
> The paper is well written and discusses the related literature well. I could not go through the details of error analysis, but the statement of the results is rigorous. I have some question/comments about the assumptions and results.
>
> Thank you very much for noticing the merits of our paper.
>
> 1- What is the motivation to study bi directional GAN as opposed to GAN? Can these results be extended to GAN, or is there a special property of GAN that allows relaxing the assumptions?
>
> The main motivation to study bidirectional GANs (BiGANs) is that it has the ability to learn an encoder, that is, BiGANs can learn a representation of the data. In the cited paper by Donahue et al. (2017), they showed that the intermediate representations are beneficial in various tasks empirically. Moreover, BiGANs make use of the joint distribution of data and latent representations, which can better capture the information of data than the vanilla GANs.
> We have some results for GANs where we can get rid of the assumption that the data distribution is absolutely continuous. However, we require this assumption for BiGANs currently. This is due to the fact that in the error analysis of BiGANs, we need to control the approximation error of the encoder network that pushes  the data distribution to the source distribution, in addition to the approximation error of the generator network.
>
> 2- What are the limitations of Assumption 2? Does it exclude data distributions that lie in low-dimensional manifolds? How necessary is this assumption?
>
> Yes, this assumption excludes the low dimensional manifold data distribution case, which is the main limitation of this assumption.  This assumption is needed in bounding the generator and the encoder approximation errors (Theorem 4.3) due to the bidirectional structure. In the original GANs, we do not need this assumption for data distribution, and the result can be extended to the low dimensional manifold data distribution case. How to explore the possible low-dimensional latent structure of data in BiGAN is an interesting problem, we will study it in the future work.
>
> 3- For Theorem 3.2., the constant for the unbounded case does not seem to be explicit as the paper suggests. It seems to better distinguish this in the introduction
>
> Great suggestion! We used two methods bounding the statistical errors, which led to two different bounds: one with an explicit square root-d prefactor with the cost that we have an additional  $\log n$ factor;  Another bound has an implicit prefactor but without the  $\log n$ factor.  The overall error bound is the minimum of these two bounds.  Hence, there is a trade-off between the explicitness of the prefactor and the $\log n$ factor. In the revision, we will clarify this point.
>
> 4- It is surprising that the dimension k does not appear in result in Theorem 3.3. Is there any explanations for this?
>
> Thank you very much for catching this. We made a mistake here: it should be $k$ instead of $1$ in the denominator, i.e., $\frac{1}{d+1}$ should be $\frac{1}{d+k}$ in the expression for the error bound! We will correct this error in the revision.
>
> 5- I did not understand how the results in Theorem 4.3 is extended to multi-dimensional case for $\nu$.
>
> Since $\nu$ and $\mu$ are absolutely continuous by assumption, they are also absolutely continuous in any one dimension. Hence the proof reduces to the one-dimensional case. We will make this clear in the revision.
>
> Overall, the paper is well written and comparison is given with the related literature. I only suggest including an outline to navigate the theoretical results
>
> Thank you very much again for your positive feedback and constructive suggestions. We will add an outline for the theoretical results in the revision.

---

> > ### Comment · Reviewer_eow5 · 2021-08-22
> > **After reading rebuttal**
> >
> > Thank you for your response to my questions and concerns. I think it is a good theoretical paper, but a bit weak on impact and implications. I increase my score to 6.

---

> > > ### Author Response · Authors · 2021-08-23
> > > **Response to your further comments**
> > >
> > > Thank you for your additional comments.
> > >
> > > We really appreciate your work in reviewing our submission and your helpful comments and constructive suggestions. We will revise our paper incorporating your comments and suggestions. We believe your review helped us improve the paper and we are grateful to you for that.

---

### Decision · Program_Chairs · 2021-09-27

**Decision:**

Accept (Poster)

**Comment:**

This paper presents non-asymptotic error analysis for the bidirectional GANs.
In the analysis the authors have succeeded in relaxing those assumptions usually adopted in the GAN literature, such that the latent distribution and the data distribution has the same dimension and that the true data distribution has a bounded support. They have also succeeded in making some of the coefficients in the error bounds explicit. These constitute the main contributions of this paper.

The reviewers raised several questions, and the authors have addressed them adequately, as evidenced by two of the reviewers having raised their scores from below the threshold to above. Now all the review scores are above the threshold, I am happy to recommend acceptance of this paper for presentation at the NeurIPS conference.

Minor points:
- A learnability issue: Lemma 4.2 is based on existence of a neural network $\\psi$, and Theorem 4.3 is based on existence of neural networks $g$ and $e$, with desired properties. These lemma and theorem do not care about the learnability of these neural networks. It would therefore be nicer if some empirical results be presented to at least suggest that the learnability issue would be minor in practice, so that the stated error bounds are indeed relevant.
- Line 34: join(t)
- Line 59: the data joint distribution.( )To the best
- Lines 93, 94: $(\sim)\nu$, $(\sim)\mu$
- Lines 121, 124: vector(s)
- Line 138: metri(ci)zing
- Equation between lines 186 and 187: $h\in(\mathcal{F}\to\mathcal{H})$
- Definitions of $\mathcal{E}_3$ and $\mathcal{E}_4$: In the proof of Lemma 4.1, it seems that the authors used slightly different definitions of these quantities, where $g^*$ and $e^*$ are replaced with $\hat{g}$ and $\hat{e}$, respectively.
- Line 267: cover(ing) numbers
- SM, line 26: $|z_0|\le\log n$ $\to$ $\\\|z_0\\\|\le\log n$